palaeontology

ophiuroid, three-dimensional imaging, locomotion

**Author for correspondence:**
E. G. Clark
e-mail: elizabeth.g.clark@yale.edu

# Three-dimensional visualization as a tool for interpreting locomotion strategies in ophiuroids from the Devonian Hunsrück Slate

E. G. Clark[1], J. R. Hutchinson[2] and D. E. G. Briggs[1,3]

[1]Department of Earth and Planetary Sciences, Yale University, 210 Whitney Avenue, New Haven, CT 06511, USA
[2]Structure and Motion Laboratory, Department of Comparative Biomedical Sciences, The Royal Veterinary College, Hawkshead Lane, Hatfield, Hertfordshire AL9 7TA, UK
[3]Yale Peabody Museum of Natural History, Yale University, New Haven, CT 06520, USA

EGC, 0000-0003-4289-6370; JRH, 0000-0002-6767-7038; DEGB, 0000-0003-0649-6417

Living brittle stars (Echinodermata: Ophiuroidea) employ a very different locomotion strategy to that of any other metazoan: five or more arms coordinate powerful strides for rapid movement across the ocean floor. This mode of locomotion is reliant on the unique morphology and arrangement of multifaceted skeletal elements and associated muscles and other soft tissues. The skeleton of many Palaeozoic ophiuroids differs markedly from that in living forms, making it difficult to infer their mode of locomotion and, therefore, to resolve the evolutionary history of locomotion in the group. Here, we present three-dimensional digital renderings of specimens of six ophiuroid taxa from the Lower Devonian Hunsrück Slate: four displaying the arm structure typical of Palaeozoic taxa (*Encrinaster roemeri*, *Euzonosoma tischbeinianum*, *Loriolaster mirabilis*, *Cheiropteraster giganteus*) and two (*Furcaster palaeozoicus*, *Ophiurina lymani*) with morphologies more similar to those in living forms. The use of three-dimensional digital visualization allows the structure of the arms of specimens of these taxa to be visualized *in situ* in the round, to our knowledge for the first time. The lack of joint interfaces necessary for musculoskeletally-driven locomotion supports the interpretation that taxa with offset ambulacrals would not be able to conduct this form of locomotion, and probably used podial walking. This approach promises new insights into the phylogeny, functional morphology and ecological role of Palaeozoic brittle stars.

# 1. Introduction

## 1.1. Investigating ophiuroid locomotion

Locomotion strategies reflect the relationship of an organism to its environment—how it feeds, reproduces, protects itself from predators and where it lives. The integrated body structure of an organism and the morphology of its components determine the locomotion strategies available to it [1]. The majority of living ophiuroids employ a unique strategy involving the coordination of whip-like motions of their five muscular arms, in contrast to their closest living relatives including the sea stars and sea urchins [2–4], which primarily use tube feet and move more slowly. Reconstructing the evolutionary history of ophiuroid locomotion is challenging as the morphology of the arms of many fossil ophiuroids differs from that in living forms, from the basic structure and number of elements to their arrangement and integration. Determining the movement capabilities and inferring the locomotion strategies of extinct ophiuroid morphologies is necessary to better understand their ecology and to reconstruct the evolutionary history of modern ophiuroid locomotion.

## 1.2. Phylogeny and classification of total group ophiuroids

Four orders of total group ophiuroids have been recognized [5]: Oegophiuroida [6], Phrynophiurida [6], Stenurida [7] and Ophiurida [8]. Oegophiuroida was thought to include one extant taxon, *Ophiocanops* [5,9], but this genus has been reinterpreted as falling within Ophiurida, family Ophiomyxidae [10–12]. Phrynophiurida is post-Palaeozoic [13,14] and originally included Euryalina and Ophiomyxidae [9]. However, the euryalids were found to be a monophyletic clade nested within the Ophiurida and the Ophiomyxidae were resolved elsewhere within the Ophiurida based on a molecular phylogenetic analysis [14]. Now, asterozoans are typically assigned to four major groups: Somasteroidea, Asteroidea (fossil and living sea stars), Ophiuroidea (fossil and living brittle stars) and Stenuroidea [15]. Oegophiuroida and Euryalida are included within Ophiuroidea [12]. The Stenuroidea and Oegophiuroida are exclusive to the Palaeozoic, whereas euryalids are post-Palaeozoic [11–14,16]. All four major asterozoan groups first appear in the Ordovician [17], but molecular clock estimates suggest that the asterozoans originated in the Cambrian [14], although no Cambrian asterozoans have been found [17].

Ophiuroids have been hypothesized to have evolved from a somasteroid ancestor [15,18]. The relationship between Stenuroidea and crown ophiuroids remains uncertain [17]. Stenuroids have been interpreted as stem-group ophiuroids that exemplify a 'Palaeozoic' morphology, i.e. one that includes most notably the presence of unfused ambulacral ossicles [19,20]. Smith *et al.* [20] combined molecular and morphological evidence to generate a phylogeny of ophiuroids incorporating two fossil taxa, and Shackleton [18] provided the most comprehensive phylogeny of Ordovician Asterozoa, illuminating relationships between several major families. Blake & Guensburg [17] presented a hypothesis of relationships between the Somasteroidea, Stenuroidea, Ophiuroidea and Asteroidea: the somasteroid *Archegonaster* was found to be nested within the stenuroids, raising questions regarding the monophyly of these groups. Although phylogenomic tools have been applied to extant taxa [12,14], there has been no comprehensive phylogenetic analysis of total group ophiuroids.

Uncertainty about ophiuroid relationships is owing, in part, to the elimination of much morphological diversity during the Permian–Triassic (P/T) extinction [21,22]. Molecular clock estimates suggest that crown-group ophiuroids originated in the Permian [14]. All Permian families of Ophiurida, except Aganasteridae, have Triassic representatives [21] and the Palaeozoic genera *Aganaster* and *Stephanoura* show similarities to the modern *Ophiomusium* and are assigned to Ophiurina [23,24]. Diversity at the genus/species level was greater in the lower Triassic than in the Permian, indicating that post-extinction recovery of ophiuroids was relatively rapid [21]. This was accompanied by the radiation of the crown group, characterized by the 'modern' arm morphology (with fused ambulacrals forming vertebrae) [19]. Thus, a transition is evident in the fossil record between the 'Palaeozoic' arm morphology, which was dominant in the Palaeozoic, and the 'modern' arm morphology that was present from the Devonian onward and diversified rapidly following the P/T extinction [21,24]. It is not known how the crown is related to the stem, nor is it clear whether or not the 'modern' morphology is an apomorphy of the crown [24], as has been suggested [19,20]. The functional consequences of the 'Palaeozoic' arm morphology have not been investigated, leaving the origin of musculoskeletally-driven brittle star locomotion unknown.

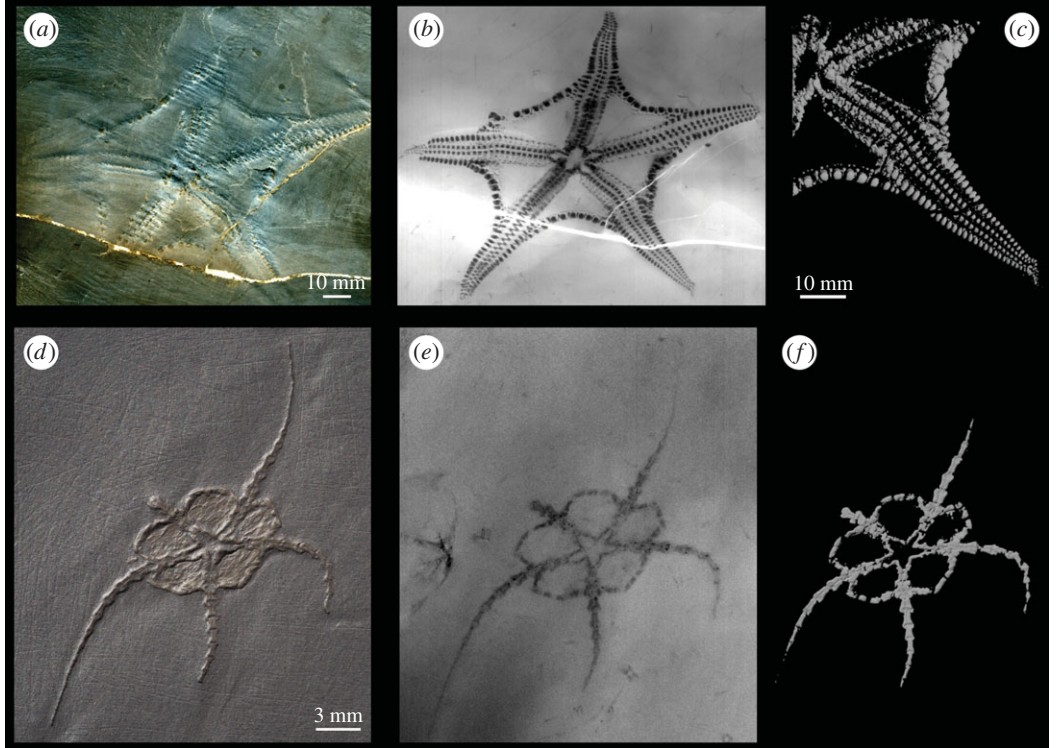

**Figure 1.** (*a–c*) *Euzonosoma tischbeinianum* (EGR 27), (*d–f*) *Ophiurina lymani* (HS 705). (*a,d*) Photograph: note that EGR 27 has undergone minimal preparation. (*b,e*) X-ray image. (*c,f*) Digital rendering of the three-dimensional micro-CT image in Autodesk Maya, (*c*) shows only one arm of EGR 27.

## 1.3. Arm morphology and movement

Living ophiuroids typically have five arms made of approximately 100 repeating segments of five ossicles each [25]. One large ossicle (the vertebra, formed from fused ambulacrals [24]) forms the centre of each segment. Living ophiuroids generally coordinate periodic musculoskeletally-driven oscillations of their arms to effect rapid locomotion [26–28]. Related taxa such as sea stars and sea urchins [3,4,14,29], by contrast, use tube feet as a primary tool for locomotion. This suggests that musculoskeletally-driven locomotion in living ophiuroids is derived from a tube foot-walking ancestor. Elucidating the functional capabilities of ophiuroid arms that lack vertebrae is key to illuminating this mechanical transition.

Motion in musculoskeletal-based organisms, and in many robots with analogous rigid structural components, relies on the application of rotational forces about a point at the interface between two skeletal structures (i.e. a joint). The skeletal structure of living ophiuroid arms comprises modular segments with a large fused vertebra at the centre [30,31]. These vertebrae accommodate the muscle attachments between successive segments and the joint interface: rotational forces generated by the muscles cause the distal segments to rotate about the joint [31].

Palaeozoic ophiuroids [19] are dominated by taxa with two columns of ambulacral ossicles along the central axis of the arm (i.e. separated by the perradial line) (figures 1–3), in contrast to the single central column of fused ossicles (vertebrae) in extant ophiuroids. The columns of ambulacrals are typically offset (alternating). Certain taxa have ambulacrals symmetrical across the midline of the arm (opposing), and phylogenetic evidence suggests that this state probably originated multiple times [18]. The morphology of the ambulacrals varies, although in many taxa, the ventral surface has been described as having a boot-shaped ridge [32–34].

Previous inferences regarding the movement capabilities and locomotion strategies of Palaeozoic ophiuroids with two rows of ambulacrals have been made based on the morphology of their arms [5,19,32]. Spencer & Wright [5], for example, suggested that the articulations and muscles in Stenurida only allowed 'simple movements' and that Oegophiuroida did 'not move speedily on the seafloor' owing to constraints imposed by their arm morphology. Glass & Blake [32] suggested that Palaeozoic ophiuroids had 'limited arm mobility' and most likely used tube foot-walking as do modern

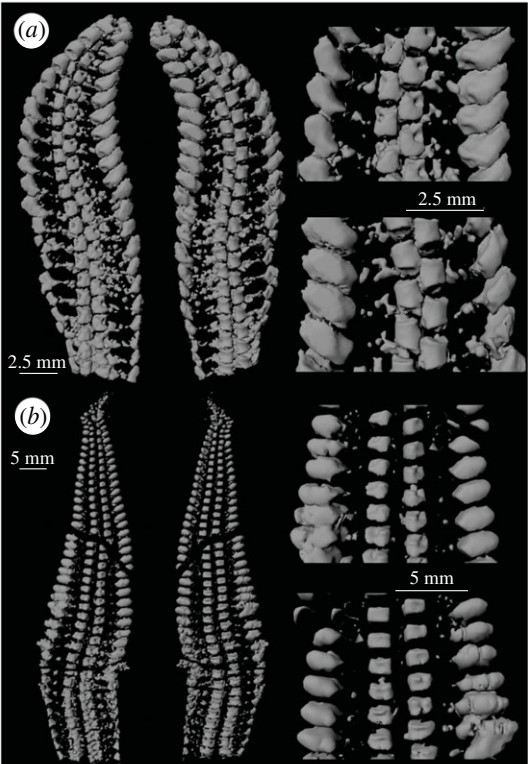

**Figure 2.** Arms of (*a*) *Encrinaster roemeri* (Hubo 116) and (*b*) *Euzonosoma tischbeinianum* (EGR 27). Views of ventral (left) and dorsal (right) surface of arm and details of rows of ossicles on the ventral (above) and dorsal (below) sides.

asterozoans. Hunter & McNamara [19] suggested that ophiuroids with vertebrae have greater mobility than those without, allowing them to radiate into a wider variety of ecological niches.

However, to form reliable inferences about the production of musculoskeletally-driven motion, both the geometry of the surfaces of the skeletal structures and their three-dimensional position with respect to one another must be understood [31]. The precise positioning between successive ambulacrals in Palaeozoic ophiuroids has been difficult to determine: although careful preparation can reveal key features, standard methods for investigating the morphology of extinct organisms (i.e. fossil preparation) present a trade-off between observation of individual elements and visualization of how they fit together. It is not possible to view the full surface area of fossilized skeletal elements directly while retaining their precise three-dimensional geometric positioning relative to one another *in situ*, particularly where they abut one another directly, because the point of articulation is obscured. Here, we used micro-computed tomography (CT) scanning and three-dimensional visualization technology to digitally remove the matrix of several fossil ophiuroid specimens, revealing the *in situ*, articulated morphology of several fossil ophiuroids in three dimensions. This technique represents the only non-destructive strategy available to document both the three-dimensional geometry and the articulations between successive elements in 360°.

Three-dimensional imaging and digital visualization have been used previously to reveal the morphology of the water vascular system of the Ordovician ophiuroid *Protasterina* from Kentucky [35] and to interpret accumulations of ophiuroids in the Lower Devonian Bokkeveld Group of South Africa [36,37]. We developed three-dimensional digital reconstructions of six Hunsrück Slate ophiuroids based on micro-CT images of representative well-preserved, fully articulated fossil specimens as a preliminary test of previous interpretations of their mode of locomotion via tube feet or musculoskeletally-driven movement. This study also provides an assessment of the potential of micro-CT scanning and digital visualization to further our understanding of Palaeozoic ophiuroids.

## 2. Material and methods

The Lower Devonian Hunsrück Slate of western Germany has yielded a remarkable diversity of pyritized fossils during roof slate mining [38,39]. We selected this locality as it has yielded a notable number of

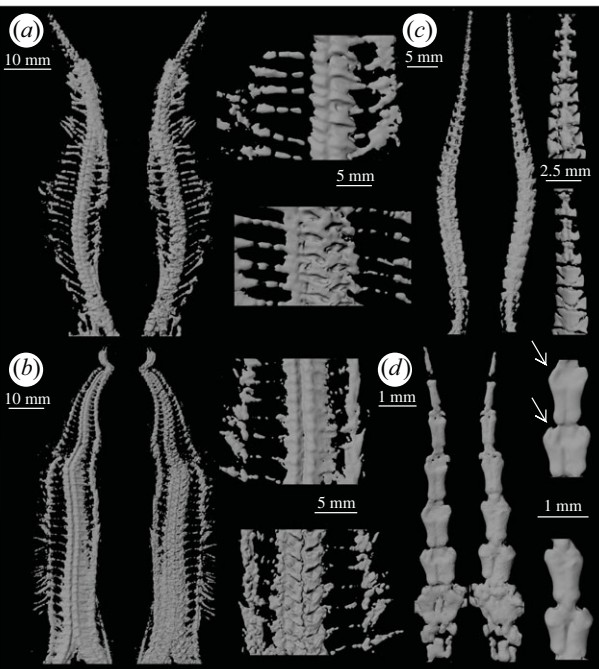

**Figure 3.** Arms of (*a*) *Loriolaster mirabilis* (ESCH 596), (*b*) *Cheiropteraster giganteus* (Hubo 119), (*c*) *Furcaster palaeozoicus* (OKL 96) and (*d*) *Ophiurina lymani* (HS 705). Views of ventral (left) and dorsal (right) surface of the arm and details of the arm ossicles on the ventral (above) and dorsal (below) sides (dorsal and ventral details of *F. palaeozoicus* and dorsal details of *L. mirabilis* are from a different arm of the same specimen). Arrows indicate depressions corresponding to podial basins in *O. lymani*.

complete Palaeozoic ophiuroids [5,40], some with soft tissue preservation including tube feet [32], as well as ophiuroid trackways [32,41]. Traditional preparation of Hunsrück Slate material involves scraping away the slate matrix with wire brushes and steel picks (figure 1*d*), which runs the risk of removing critical details from the specimens. Careful airbrasive preparation with iron powder retains spectacular detail, but does not reveal the specimen in the round [32,38]. As micro-CT imaging illuminates density contrasts, this technique is a useful tool for visualizing fossil specimens where there is a strong density differential between the material comprising the fossil and the surrounding matrix, or where preservation is mouldic [42]. In the case of the Hunsrück Slate, three-dimensional image processing tools can be used for the digital removal of the slate matrix from the pyritized fossils (figure 1). This technique also illuminates the morphology in three dimensions, including internal structures in some cases, and allows for easy data dissemination [43].

This investigation was conducted using the extensive Hunsrück Slate ophiuroid material at the Goldfuß Museum of the Steinmann Institute, which houses an X-ray and micro-CT scanning facility at the University of Bonn. An initial selection of specimens for micro-CT scanning excluded those too large to be accommodated in the scanner and those in which ossicles had separated because of decay or where surface detail had been lost as a result of previous preparation. Thirty-three specimens which promised at least one completely articulated arm were X-rayed (e.g. figure 1*b,e*) to evaluate the degree of detail preserved. Many of these articulated specimens were too lightly pyritized or overgrown with pyrite internally even though they appeared pristine on the surface. Diagenetic factors leading to merging, compaction or shearing of skeletal elements eliminated several others. In this way, we determined the best specimens for micro-CT scanning in which the morphology of individual ossicles (or rows of ossicles) was evident in the X-ray image.

We acquired digital images of one arm of each of six specimens, representing six species in four families: Encrinasteridae—*Encrinaster roemeri* (Hubo 116), *Euzonosoma tischbeinianum* (EGR 27) [18,33]; Cheiropterasteridae—*Loriolaster mirabilis* (ESCH 596), *Cheiropteraster giganteus* (Hubo 119) [44]; Furcasteridae—*Furcaster palaeozoicus* (OKL 96) ([33,45,46]; note that *Furcaster* was previously assigned to Protasteridae [7]) and Ophiurinidae—*Ophiurina lymani* (HS 705) [5] (figures 2 and 3).

The extent of mechanical preparation varies between the chosen specimens (figure 1); in each case, however, the morphology is largely obscured by the slate matrix ensuring minimal damage to the morphology. Micro-CT scanning allowed for visualization of the three-dimensional morphology and

*in situ* positioning of the skeletal elements through 360°, and revealed the nature of the contact between adjacent arm ossicles [31].

The specimens were imaged using the GE phoenix|x-ray v|tome|x micro-CT scanner at the Steinmann Institute, University of Bonn and visualized using VG STUDIO MAX in the Briggs and Bhullar laboratories at Yale University. Ossicles were extracted from the CT scan as watertight polygon meshes using VG STUDIO 3.0. Meshes of individual ossicles or rows of ossicles were cleaned in MESHLAB [47]. Skeletal features were extracted from the scans and imported for visualization in MAYA (Autodesk). These techniques were adapted from strategies for imaging and visualization in extant echinoderms [31,35,43]. The present investigation represents, to our knowledge, the first application of this approach to ophiuroids from the Hunsrück Slate.

# 3. Results

In the following descriptions of the specimens scanned, we have adopted the terminology used by the authors in [32] and [48].

## 3.1. *Encrinaster roemeri* [Encrinasteridae] (Hubo 116)

The arm of *Encrinaster roemeri* comprises two major components: ambulacrals and adambulacrals (figure 2*a*). These ossicles are aligned within a plane so that the arm appears relatively flat when visualized digitally. There are no dorsal or ventral ossicles, in contrast to the arms of the majority of extant ophiuroids [1]. Nearly one-third of the length of the arm is incorporated into the disc [40].

The ambulacrals form two columns offset along the long axis of the arm [40,49,50]. The ventral ambulacrals have a boot-shaped ridge: the length of the 'leg' and the width of the 'foot' are approximately equal, and the length of the 'toe' is approximately half the length of the leg (terminology in [32, fig. 6]). The width of the 'distal fitting' and 'centre leg' are approximately equal to the length of the toe. The dorsal surface is trapezoidal; the lateral margin is shorter than the central margin. There are slight mediolateral ridges towards the proximal and distal ends of the dorsal surface. The proximal and distal faces are relatively flat with several ridges which may represent sites of muscle attachment. The axial face is relatively flat. The 'lace area' [32] flanks a depression which presumably represents a podial basin hosting a tube foot as in the Ordovician ophiuroid *Protasterina flexuosa* [35]. There are spaces between successive ambulacrals within the same column which can be visualized in the micro-CT scan but are obscured in the fossil. There appears to be a gap along the midline of the arm between the columns of ambulacrals, but the digitized specimen shows that the ossicles fit together tightly: there is little or no space between the columns. There is a reduction in the width of the distal fitting and the size of the podial basin relative to other features of the ambulacrals along the length of the arm.

The adambulacrals are relatively simple curved structures, slightly convex abradially [40]. The long axis of the adambulacrals is positioned at approximately 40° to the axis of the arm. The adambulacrals of the free arm are larger than those within the disk. Pyritized elements extend between the ventral margin of the adambulacrals and the toe of the ambulacrals.

## 3.2. *Euzonosoma tischbeinianum* [Encrinasteridae] (EGR 27)

The ambulacrals and adambulacrals are arranged in two columns offset across the arm axis, as in *Encrinaster roemeri* (figures 1*a–c* and 2*b*) [40]. Ventral and dorsal ossicles are absent. More than half of the length of the arm projects beyond the disk. The degree of separation of the two columns varies between arms, presumably as a result of taphonomic processes.

The ambulacrals have a boot-shaped ridge on the ventral surface. The length of the leg is shorter than the width of the foot along most of the arm, but they approach the same length towards the arm extremity. The length of the toe is shorter than the width of the distal fitting and less than half the width of the foot. The width of the distal fitting and centre leg are approximately equal. The dorsal surface of the ambulacrals is roughly rectangular in outline. Flared proximal and distal ridges are associated with a depression along the mediolateral axis of the dorsal face of the proximal ossicles; the dorsal surface becomes gradually smoother in more distal ossicles. The proximal and distal faces are relatively flat, with ridges resembling those in *E. roemeri*. The axial face of the ambulacrals is relatively flat, with a curved depression in the centre. The most striking differences between *E. tischbeinianum* and *E. roemeri* are the greater dorsoventral thickness of the ambulacrals in *E. tischbeinianum* and the reduction in relative size of

the ambulacrals and adambulacrals towards the distal end of the arm (figure 2*a,b*). There are spaces between successive ambulacrals in *E. tischbeinianum* as in *E. roemeri* but, in contrast with *E. roemeri*, there are large gaps between the two columns of ambulacrals in this specimen.

The adambulacrals of *E. tischbeinianum* are relatively flat on both proximal and distal surfaces. The dorsal, ventral and lateral surfaces are concave. The adaxial margin is V-shaped, the distal arm of the 'V' slightly longer than the proximal. The adambulacrals become smaller along the free arm distally. They are wider than long within the free arm, in contrast to those in the fixed arm.

Descriptions based on surface preparations illustrated the adaxial margin of the adambulacrals directly abutting or even overlapping the ventral part of the ambulacrals in *Encrinaster grayi* from the Ordovician of Girvan, Scotland (e.g. [49, p. 422]). However, we observed pyritized needle-like structures between the ambulacrals and the adambulacrals in *E. roemeri* and *E. tischbeinianum* (Hubo 116 and EGR 27, figure 2*a,b*). These pyritized elements vary in morphology in each row as well as among the other arms that we X-rayed.

Jell & Theron [51, p. 165] synonymized *Euzonosoma* with *Encrinaster* (also [34]). They noted that Spencer [49] distinguished these genera on the basis of a single character: the mediolateral width of the adambulacrals. In *Euzonosoma*, there is a distinct difference between the width of these ossicles at the tip and at the base of the free arm (figure 2*b*). In *Encrinaster*, by contrast, the width is similar along the length of the free arm (figure 2*a*). Jell & Theron [51] argued that this feature reflects the angle at which these ossicles are preserved and exposed in the specimens. In the three-dimensional image of the specimens, however, it is apparent that the relative size of the adambulacrals within the free arm differs between the two taxa in line with the original designation [49]. However, other species assigned to these genera, as well as specimens representing a range of sizes, should be examined using three-dimensional imaging to determine whether the differences in ossicle shape are taphonomic, or major shape differences are consistent within these genera. In the meantime, we retain the genus *Euzonosoma* here (also [18,52]).

## 3.3. *Loriolaster mirabilis* [Cheiropterasteridae] (ESCH 596)

The ambulacrals are offset (figure 3*a*) [40,44]. The ventral surface is exposed in the specimen, and the outline of the ventral ambulacrals is not sharp in the digital image presumably owing to overpreparation as this lack of resolution is only evident where the specimen has been prepared. Ambulacrals are rectangular to trapezoidal in dorsal view [40,53]. The ambulacrals become narrower along the arm distally. Ambulacrals form an archway along the proximodistal axis where they abut, presumably housing the radial canal of the water vascular system. The proximal and distal faces of the ambulacrals appear to be relatively flat. This specimen is unusual in having only four arms ([40, Pl. 9, figs 1 and 3]), but the morphology of the elements is typical of the taxon. There is no evidence that a fifth arm was in the process of regeneration.

## 3.4. *Cheiropteraster giganteus* [Cheiropterasteridae] (Hubo 119)

The ambulacrals are offset [44,53] with a straight proximal ridge and u-shaped distal ridge in dorsal view (figure 3*b*). The width of the ambulacrals tapers abaxially. The distal face has a central depression, whereas the proximal face is relatively flat, angled at approximately 30° to the mediolateral axis. A central depression in the axial surface creates a slightly sinusoidal space for the radial canal between the paired ambulacrals [44]. The morphology of individual ossicles on the ventral surface is difficult to discern in the three-dimensional image, as is part of the area between rows, owing to inadequate preservation. The adambulacrals appear to be relatively small and slightly curved and elongate; long spines are preserved projecting from many of them [53].

## 3.5. *Furcaster palaeozoicus* [Furcasteridae] (OKL 96)

The paired ambulacrals are opposing and roughly rectangular in ventral outline [54] (longest along the proximodistal axis of the arm) with ridges on the proximal and distal faces (figure 3*c*) [44]. The ambulacrals are narrowest at their mid-length and widest towards the proximal end. The pairs are triangular in dorsal outline in the proximal part of the arm, wide distally and tapering towards the proximal end; they are approximately rectangular in the distal part. The distal face is relatively flat with a median projection similar to that in modern ophiuroids (see [31, figs 2, 6–8]), which has been interpreted as a ball-and-socket joint in *Furcaster trepidans* ([18], see also [55,56]). The proximal and

distal ridges surrounding the projection form an area on the proximal and distal faces which resembles the four muscle attachment sites in modern ophiuroids (see [31, fig. 2]). The adambulacrals are curved, often with a ventral projection on the proximal abaxial end. The width of each row of ossicles diminishes along the length of the arm.

### 3.6. *Ophiurina lymani* [Ophiurinidae] (HS 705)

Although pyritization obscures the boundaries between individual plates in this specimen, the extracted three-dimensional rendered images show the morphology of the rows of ossicles (figures 1*d*–*f* and 3*d*) [54]. There are fewer rows of ossicles than in the arms of the other Palaeozoic ophiuroids investigated here. The rows of ossicles are roughly rectangular in both dorsal and ventral view, longest in the proximodistal axis. A median junction runs along the axis of the arm and is evident on both the dorsal and ventral surfaces. The distal face appears slightly convex and abuts the concave face of the next row. Rows of ossicles widen distally (excluding the most distal row). The proximal rows are wider than those at the tip of the arm.

## 4. Discussion

### 4.1. Inferring locomotion strategies in fossil asterozoans

Analysing function in fossil animals usually relies on adapting and applying techniques used to study function in living forms (e.g. [57]). The arms of living ophiuroids comprise modular segments, each with a single central vertebral ossicle, which is bilaterally symmetrical along the axis of the arm. These vertebrae are connected by ball-and-socket articulations (joint interfaces) which are surrounded by muscles. Contraction of the muscles flexes the distal segment about the joint [31]. Such joint interfaces are integral to performing musculoskeletally-driven locomotion in living animals [31].

Taxa such as those examined here with two offset rows of ambulacrals and arms largely incorporated into the disc have been interpreted as walking with their tube feet, as opposed to using musculoskeletally-driven motion [5,32,58]. In this study, digital visualization of the best-preserved specimens with two offset rows of ambulacrals (*Encrinaster roemeri*, *Euzonosoma tischbeinianum*) revealed a lack of joint interfaces between successive elements within columns of ambulacrals, indicating that they could not have performed musculoskeletally-driven locomotion as in living ophiuroids. The ambulacrals in these ophiuroids tend to be separated proximodistally and, in some cases, tightly juxtaposed across the midline of the arm (in our specimens of *E. roemeri*, *L. mirabilis*). The closest living relatives of ophiuroids (Asteroidea, Echinoidea) use podial walking for locomotion [2–4] and the musculoskeletally-driven locomotion strategy of extant ophiuroids is derived. This suggests that the unique locomotion strategy used by modern ophiuroids evolved after their most recent common ancestor with other modern taxa. The Palaeozoic ophiuroids examined here that lack vertebrae (Encrinasteridae, Cheiropterasteridae) thus probably relied on their tube feet for locomotion (podial walking) [32]. The formation of joint interfaces, critical to musculoskeletally-driven locomotion in extant ophiuroids, represents a necessary event in the evolution of this form of movement. Rotational forces in modern ophiuroids are applied about the joint interface located on the central axis of the vertebrae [31], suggesting that opposition and fusion of the ambulacrals to form vertebrae were key steps towards musculoskeletally-driven locomotion.

The arrangement of ambulacrals in *F. palaeozoicus* (Furcasteridae), among the ophiuroids examined here, is the most similar to that in extant taxa (figure 3*c*). The rows of ossicles are bilaterally symmetrical. Some rows appear to preserve ball-and-socket joints, resembling the structure and location of joint interfaces and muscle attachment sites in extant ophiuroids, suggesting that musculoskeletally-driven locomotion may have been possible. The relative length of the free versus the fixed arm is much greater in *F. palaeozoicus* than in the other taxa examined here; the long, flexible arms (e.g. [39, fig. 81]) unencumbered by disc tissue are consistent with musculoskeletally-driven motion as is the arm morphology. Gladwell [48] considered that the flexible arms of the Silurian species *Furcaster leptosoma* suggest high mobility and that the large tube feet may have aided in locomotion as well as feeding.

Spencer & Wright [5] considered whether some Devonian Ophiurida (designated 'modern' by Hunter & McNamara [19]) may have been capable of active movement owing to their seemingly flexible arms. The morphology of the rows of arm ossicles of *O. lymani* (Ophiurinidae) (figures 1*d*–*f* and 3*d*), however, is remarkably similar in shape to those of *Amphicutis stygobita* [59, figs 3,6,7,10, 60], an extant cave-dwelling brittle star which locomotes mainly by tube foot walking. The rows of arm ossicles in both are bilaterally symmetrical and elongate, widening distally. Distal widening in *O. lymani* accommodates structures that

we interpret as podial basins (figure 3*d*), similar in scale to those in *A. stygobita*. The extent to which such similarity can be relied on as evidence for mode of locomotion, however, is uncertain. Living ophiuroids, such as Ophiomusaidae and Ophiosphalmidae [61,62], with arm structures that show similarities to that of *O. lymani*, use musculoskeletally-driven locomotion. Further analyses are needed (as in [31]) to understand the relationship between structure and function in these forms.

Three-dimensional digital imaging is the only non-destructive strategy currently available that allows simultaneous visualization of the three-dimensional geometry of the skeletal elements of fossil ophiuroids and their relative positioning *in situ*. This information is a necessary basis for biomechanical inferences such as the geometry of the joint interfaces and the range of motion [31,42]. Application of the methodology outlined here to a broad scope of Palaeozoic ophiuroids with opposing and fused ambulacrals (such as *Eospondylus* [11]) is necessary to illuminate the evolutionary history and ecological context of the rise to dominance of musculoskeletally-driven locomotion in brittle stars. Our open three-dimensional morphological dataset of the six study specimens render this application, and others, available to future researchers.

# 5. Conclusion

Prior to this study, the three-dimensional morphology of the arm ossicles of Hunsrück Slate ophiuroids was mainly inferred by integrating the morphology of surfaces exposed by preparation, isolated arm ossicles and some additional information from X-radiographs. The non-destructive approach used in our investigation revealed the full morphology of the arm skeleton and the relative position between successive ossicles in Hunsrück Slate ophiuroids, to our knowledge for the first time *in situ*, limited only by the quality of preservation. This approach, which has also been applied to living ophiuroids [31], allows for the most complete three-dimensional visualization of their morphology available to date. Micro-CT imaging also has the potential to be advantageous for visualization of skeletal elements in the disc. The three-dimensional imaging data support the hypothesis that the 'Palaeozoic' ophiuroid specimens examined here with alternating ambulacrals used podial walking, as structures necessary for musculoskeletally-driven locomotion as in modern ophiuroids were absent. The paired ambulacrals and ball-and-socket joints in *F. palaeozoicus*, together with their long free arms, suggest that musculoskeletally-driven motion may have been possible. Fusion of opposed vertebrae and the creation of joints between successive rows represent critical transitional steps in the origin of modern brittle star musculoskeletally-driven locomotion. Future study of Palaeozoic brittle stars integrating three-dimensional imaging with dynamic modelling may provide further insights into the evolution of locomotion in ophiuroids.

Ethics. We used no live animals in this study. The fossil specimens are housed in the Goldfuß Museum of the Steinmann Institute of the University of Bonn and were not collected as part of this investigation.

Data accessibility. Fossil specimens used in this investigation (Hubo 116, EGR 27, ESCH 596, Hubo 119, OKL 96, HS 705) are housed at the Goldfuß Museum of the Steinmann Institute of the University of Bonn. The micro-CT scans are available from FigShare: https://doi.org/10.6084/m9.figshare.11886573.v1.

Authors' contributions. E.G.C., J.R.H. and D.E.G.B. were involved in conceptualization and funding acquisition. E.G.C. processed the micro-CT data. E.G.C. and J.R.H. constructed the digital models. E.G.C., J.R.H. and D.E.G.B. interpreted the results and prepared the manuscript.

Competing interests. We have no competing interests.

Funding. This project was funded by the National Science Foundation (NSF Award 1701830), the Yale Institute for Biospheric Studies, the Paleontological Society, the Silliman College George Shultz Fellowship and the Pierson College Richter Fellowship.

Acknowledgements. We are grateful to Alex Glass (Duke University), Fred Hotchkiss (Marine and Paleobiological Research Institute), Christoph Bartels, Gabriele Kühl, Jes Rust, Teresa Franke, Alexandra Bergmann, Georg Oleschinski (University of Bonn), Bhart-Anjan Bhullar, the Bhullar Laboratory, the Briggs Laboratory, Travis Brady, Sloane Smith (Yale University), Eva Herbst, Krijn Michel, Andrew Cuff, Peter Bishop and Louise Kermode (Royal Veterinary College) for advice and assistance. We acknowledge the roof slate miners who discovered many of the Hunsrück Slate fossil specimens. We are very grateful to the reviewers and editors (Jeffrey Thompson, Kevin Padian) whose detailed comments enabled us to improve the paper significantly.

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
