## [Reviewer comments · Royal Society Open Science]

Review History

RSOS-200329.R0 (Original submission)

Review form: Reviewer 1

Is the manuscript scientifically sound in its present form?

Yes

Are the interpretations and conclusions justified by the results?

No

Is the language acceptable?

Yes

Do you have any ethical concerns with this paper?

No

Have you any concerns about statistical analyses in this paper?

No

Recommendation?

Accept with minor revision (please list in comments)

Comments to the Author(s)

Dear authors,

It was my pleasure to review your article and I would like to congratulate you on this interesting piece of work. You did an excellent job in selecting appropriate specimens and in producing 3D digital renderings of some Paleozoic ophiuroids. However, I have a few concerns, especially regarding the conclusions of your paper.

First of all, there is a considerably diversity of morphological terminologies used for Paleozoic ophiuroids. I saw that you cited references in some parts of your descriptions. For a paper that focuses on function and morphology, however, it would be more advisable to pick an existing terminology, preferably one that makes sense in the context of a functional interpretation of the skeleton, and provide the relevant references in the materials and methods section. Alternatively, if you feel the existing terms preclude a meaningful interpretation of function, you could come up with a new terminology. Either way, there is a general need of clarification in this respect.

Regarding the conclusions of your paper, I agree with you that there are fundamental differences between the taxa you analysed in terms of arm structure. I also agree that these differences most probably reflect contrasting locomotion modes. However, I cannot see in what way the 3D digital renderings provide new insights. I do not question the interest of such approaches but I feel their surplus evidence is not conveyed in a convincing way here. For example, it was clear from previous works that ophiuroids with alternating ambulacrals most probably employed podial walking. You even cite references that drew that conclusion from their own morphological observations, so I cannot see in what way your data provide new insights here.

Also, you quite correctly stress that many critical details are undetectable when observing articulated skeletons in dorsal and ventral views only. Reports of dissociated skeletal parts (e.g. Boczarowski 2001), however, provide exactly that type of detail, e.g. presence or absence of ball-and-socket articulations. So again, I cannot see how your data provide new insights here.

Then I see a fundamental problem with conclusion in lines 340 - 342: how can you say that "the expression of musculoskeletally driven locomotion occurred after the evolution of opposing ambulacrals, but preceded the fusion of the ambulacrals to form vertebrae" when you analyse only taxa with ambulacrals either alternating or fused into vertebrae? Furcaster has the ambulacral halves not only opposing but firmly fused into vertebrae, much like the living ophiuroids. It differs fundamentally in articulation type and in the position of the water vessel canal, and I would have loved to see if these differences matter in locomotion.

As for *Ophiurina lymani*, I do agree that there are some similarities with *Amphicutis stygobita* but there are many other living species with similar arm morphologies that do not rely on podial walking (e.g. most ophiomusaid and ophiosphalmid taxa). The position of the podial basins is not unusual at all since almost all ophiuroids have their tube feet at the distal ends of the arm segments.

Altogether, I feel that the evidence you provide is surely interesting but fails to provide convincing new evidence to draw conclusions on the mode of locomotion of the analysed species. Much more morphological evidence is needed and many more morphological comparisons have to be made before any kind of generalised conclusion can be drawn.

I definitely recommend publication of the manuscript but with a much more cautious approach to the conclusions.

Review form: Reviewer 2

Is the manuscript scientifically sound in its present form?

No

Are the interpretations and conclusions justified by the results?

No

Is the language acceptable?

Yes

Do you have any ethical concerns with this paper?

No

Have you any concerns about statistical analyses in this paper?

No

Recommendation?

Reject

Comments to the Author(s)

Review of Three-dimensional morphology and locomotion of ophiuroids from the Devonian. Hunsrück Slate by E. G. Clark, J. R. Hutchinson, D. E. G. Briggs. It is true that Living brittle stars employ a very different locomotion strategy and that Palaeozoic ophiuroids do differ markedly from that in living forms, making it difficult to infer their mode of locomotion. However, with regret, I do think this MS does not provide the empirical and experimental evidence to support much the claims. There are lots of great observations but there is a lack of hard data to back them up. As is sadly often the case using CT reconstruction data in the past, they have created amazing reconstructions that sadly only reveal data and observations already known. The MS fall short in lots of areas. It is over focussed on one site the pyritized Hunsrück Slate brittle stars and largely does not consider data from what they term morphological studies to show how our view of the form and function of these ophiuroids has really changed with the CT analysis. There really is no problem with the choice of taxa, the application of the methods which are the standards I expect of the senior author but my concerns are with the Introduction and Discussion. I think the study has been oversold in the introduction and the discussion does not meet my expectations.

I am not convinced that the 3D digital renderings really do improve our understanding of the group. This is a limited survey of just five 'archaic' ophiuroid taxa. The paper claims to have revealed the structure of the arms of these species in 3D for the first time. However we have already known the structure of the arms for over a century (Example Spencer 1914-1920). There are plenty of fossils from classic localities from the Ordovician to the Permian that clearly show 3D preservation of both surfaces. The morphological information has already been collected using traditional methods such as studying latex casts.

Introduction

Line 59: I afraid this is not the case, this might be true for Hunsrück Slate but not for the vast majority of specimens in museum collections. To say that their methods reveals both dorsal and ventral surfaces of each specimen for the first time together with other previously unobserved characters such as the proximal and distal surfaces of the arm segments is not true.

Line 129: These are in effect morphological descriptions that use classification sensu Glass & Blake without mentioning any of the background of the terminology used. In fact, at best these are half way house morphological descriptions and CT scans. It read more like a brief guide to taxa. It does not include background from existing extensive literature on these taxa by Spencer,

Blake or Glass. The descriptions have lots of basic errors that demonstrate a general ignorance of the systematics of the group. In addition there is no discussion on the influence of taphonomy on the morphology of these ophiuroids. Recent papers on the fauna of South Africa have not been included.

Discussion

Line 304: The discussion simply repeats observations that have been known for a long time and can be observed in Spencer, Blake or Glass. For we know that “ossicles of many Palaeozoic ‘archaic’ ophiuroids, are paired and offset along the axis of the arm”. It is also well known that the arms are largely incorporated into the body disc. The observation that “there are no extant animals that use two offset columns of repeating segments for musculoskeletally driven locomotion is very interesting but I don’t understand from the MS how this influences movement.

Line 313: It is a very bold claim to say that archaic ophiuroids (Encrinasteridae, Cheiropterasteridae) did not use musculoskeletally driven locomotion similar to that in modern ophiuroids. So and they must have relied on their tube feet for locomotion (‘podial walking’). I don’t see the logic behind this assumption. Why could they have moved their arms? In addition, they have not taken into account that many parts of the ophiuroid skeleton are decalcified. The authors have not taken into account the soft tissue structures that could have existed. It is interesting that the more derived *Furcaster palaeozoicus* among the Palaeozoic ‘archaic’ ophiuroids is the most similar to that in typical extant taxa with bilaterally symmetrical segments, ball-and-socket joints, resembling the structure and location of these joint interfaces and muscle attachment sites. However this is rather a large jump in logic without a functional test of how both these arms functioned. This makes the discussion highly speculative.

Line 326: Why does the morphology of the arm segments of *Ophiurina lymani* provide compelling evidence for podial walking? It is a bit of a jump in logic to say that they are remarkably similar in shape to those of *Amphicutus stygobita*, the extant cave-dwelling brittle star and thus must be a tube feet walker. So many other ophiuroids have this morphology and are not tube foot walkers!

Line 332: The most important result of this study is the evidence of disparity of locomotory function in the pyritized Hunsrück Slate brittle stars. However as warning, details of the articular surface of the vertebrae of living brittle stars do not necessarily correspond to differences in locomotory capability. I think it is true that examining 3D morphology reveals striking contrasts in the morphology of the arms of Encrinasteridae + Cheiropterasteridae, *Furcaster palaeozoicus* and *Ophiurina lymani*, which imply significant differences in function. It is clear there is an evolutionary sequence of development in the arms moving from stepped ambulacrals and inline ambulacrals which is very interesting. But why did you choose the Hunsrück Slate brittle stars when this transformation was very likely to have taken place by the late Silurian? This is in mind it is simply not possible to state that musculoskeletally driven locomotion occurred after the evolution of opposing ambulacrals, but preceded the fusion of the ambulacrals to form vertebrae. or that rows and columns of opposing ambulacrals must have been sufficiently articulated to exhibit this form of motion when you do not have enough data (only 5 taxa) to test this transformation, which might have already taken place by the Devonian.

Conclusions

Line 357: Finally it’s not correct to say “Prior to this study, the 3D morphology of Palaeozoic brittle star arm ossicles was mainly inferred by integrating the morphology of their dorsal and ventral surfaces exposed by preparation of different specimens.” Such information is already available from many other well preserved specimens using traditional methods. I look forward to your updated resubmitted manuscript with more convincing evidence and data.

Review form: Reviewer 3

Is the manuscript scientifically sound in its present form?

No

Are the interpretations and conclusions justified by the results?

No

Is the language acceptable?

Yes

Do you have any ethical concerns with this paper?

No

Have you any concerns about statistical analyses in this paper?

No

Recommendation?

Reject

Comments to the Author(s)

See my in text comments (Appendix A).

Decision letter (RSOS-200329.R0)

Dear Dr Clark,

Manuscript ID RSOS-200329 entitled "Three-dimensional morphology and locomotion of ophiuroids from the Devonian Hunsrück Slate" which you submitted to Royal Society Open Science, has been reviewed. The comments from reviewers are included at the bottom of this letter.

In view of the criticisms of the reviewers, the manuscript has been rejected in its current form. However, a new manuscript may be submitted which takes into consideration these comments.

Please note that resubmitting your manuscript does not guarantee eventual acceptance, and that your resubmission will be subject to peer review before a decision is made.

Your resubmitted manuscript should be submitted by 15-Oct-2020. If you are unable to submit by this date please contact the Editorial Office.

Best regards,
Lianne Parkhouse
Editorial Coordinator
Royal Society Open Science.
openscience@royalsociety.org

on behalf of Dr Jeffrey Thompson (Associate Editor) and Kevin Padian (Subject Editor)
openscience@royalsociety.org

Subject Editor Comments to Author:

Thanks for your submission. The reviewers and AE welcomed the very nice images but do not seem to think that the study brings much in the way of novel results. They also identify problems with citing relevant literature and the validity of the inferences. I agree with them that a resubmitted manuscript would need to be very different in scope and findings. It also seems that it needs a clearer expression of what exactly is new here and how this analysis improves our understanding of the functional morphology, if not the skeletal anatomy alone. I hope the reviewers' comments will be useful, and I wish you the best in reworking your study. We would consider a resubmission if it clearly answers their concerns.

Associate Editor Comments to Author (Dr Jeffrey Thompson):

Dear Elizabeth et al.,

First of all I would like to apologize for the delay in getting this back to you. With the current situation, there was a delay both in receiving reviewer comments back from reviewers, and making the associate editorial decision (this is on me).

Following the suggestions and comments of the three reviewers, I sadly feel that this manuscript isn't publishable in Royal Society Open Science in its current form. There was a general sense among the reviewers that the data presented in this manuscript, while of excellent quality (and I agree the reconstructions are stunning!), do not support the interpretations made in the discussion, and do not deliver upon the implications promised in the introduction and abstract of the paper. Unfortunately, 2/3 reviewers recommended rejection, and all three reviewers commented on the fact that, while the case is made in the manuscript that micro-ct visualization of these fossils reveals new details, that many of these morphological interpretations can be made (and have already been made) using more traditional methods. The observations made in the manuscript are thus not particularly novel. The reviewers also noted that there is a lack of appropriate attribution to previous literature (I suspect this may have to do with space limitations imposed from previous submission to Proceedings B), much of which has made similar claims to those made regarding ophiuroid morphology and mobility in the manuscript, and that there is a disconnect from much of the rest of the literature on non-Hunsrück Palaeozoic ophiuroids. I am leaving open the possibility of resubmission, following consideration of the reviewers' comments, in particular the addition of more analytical functional analyses of the morphology (a la Clark et al. 2018) might help to support the arguments made in the paper regarding arm function and mobility, as well as helping to re-frame the paper. In particular, I urge you to pay close attention to the comments of Reviewer # 2, who offered lots of details for further discussion, interpretation, and citation of previous work. In this reviewer's review, all observations deemed too general, or those having been made by previous authors, are highlighted in red, which I hope helps in re-structuring/ addressing their comments. If you do choose to re-submit to Royal Society Open Science, then I also recommend you take advantage of

our lack of length restrictions, to bolster discussion of ophiuroid morphology and reference the work and ideas of previous authors as recommended by the reviewers.

In summary, I think that given the way that the manuscript is currently written, in order to be suitable for publication in Royal Society Open Science, it would need to be overhauled in such a way that the interpretations were more clearly supported by the data, which would likely involve additional further functional/experimental analyses, and would need to be more properly contextualized within the current state of knowledge of hypotheses concerning fossil ophiuroid function.

All the best,
Jeff Thompson

Reviewers' Comments to Author:

Reviewer: 1
Comments to the Author(s)

Dear authors,

it was my pleasure to review your article and I would like to congratulate you on this interesting piece of work. You did an excellent job in selecting appropriate specimens and in producing 3D digital renderings of some Paleozoic ophiuroids. However, I have a few concerns, especially regarding the conclusions of your paper.

First of all, there is a considerably diversity of morphological terminologies used for Paleozoic ophiuroids. I saw that you cited references in some parts of your descriptions. For a paper that focuses on function and morphology, however, it would be more advisable to pick an existing terminology, preferably one that makes sense in the context of a functional interpretation of the skeleton, and provide the relevant references in the materials and methods section. Alternatively, if you feel the existing terms preclude a meaningful interpretation of function, you could come up with a new terminology. Either way, there is a general need of clarification in this respect.

Regarding the conclusions of your paper, I agree with you that there are fundamental differences between the taxa you analysed in terms of arm structure. I also agree that these differences most probably reflect contrasting locomotion modes. However, I cannot see in what way the 3D digital renderings provide new insights. I do not question the interest of such approaches but I feel their surplus evidence is not conveyed in a convincing way here. For example, it was clear from previous works that ophiuroids with alternating ambulacrals most probably employed podial walking. You even cite references that drew that conclusion from their own morphological observations, so I cannot see in what way your data provide new insights here.

Also, you quite correctly stress that many critical details are undetectable when observing articulated skeletons in dorsal and ventral views only. Reports of dissociated skeletal parts (e.g. Boczarowski 2001), however, provide exactly that type of detail, e.g. presence or absence of ball-and-socket articulations. So again, I cannot see how your data provide new insights here.

Then I see a fundamental problem with conclusion in lines 340 - 342: how can you say that "the expression of musculoskeletally driven locomotion occurred after the evolution of opposing ambulacrals, but preceded the fusion of the ambulacrals to form vertebrae" when you analyse only taxa with ambulacrals either alternating or fused into vertebrae? Furcaster has the ambulacral halves not only opposing but firmly fused into vertebrae, much like the living ophiuroids. It differs fundamentally in articulation type and in the position of the water vessel canal, and I would have loved to see if these differences matter in locomotion.

As for *Ophiurina lymani*, I do agree that there are some similarities with *Amphicutis stygobita* but there are many other living species with similar arm morphologies that do not rely on podial walking (e.g. most ophiomusaid and ophiosphalmid taxa). The position of the podial basins is not unusual at all since almost all ophiuroids have their tube feet at the distal ends of the arm segments.

Altogether, I feel that the evidence you provide is surely interesting but fails to provide convincing new evidence to draw conclusions on the mode of locomotion of the analysed species. Much more morphological evidence is needed and many more morphological comparisons have to be made before any kind of generalised conclusion can be drawn.

I definitely recommend publication of the manuscript but with a much more cautious approach to the conclusions.

Reviewer: 2

Comments to the Author(s)

Review of Three-dimensional morphology and locomotion of ophiuroids from the Devonian. Hunsrück Slate by E. G. Clark, J. R. Hutchinson, D. E. G. Briggs. It is true that Living brittle stars employ a very different locomotion strategy and that Palaeozoic ophiuroids do differ markedly from that in living forms, making it difficult to infer their mode of locomotion. However, with regret, I do think this MS does not provide the empirical and experimental evidence to support much the claims. There are lots of great observations but there is a lack of hard data to back them up. As is sadly often the case using CT reconstruction data in the past, they have created amazing reconstructions that sadly only reveal data and observations already known. The MS fall short in lots of areas. It is over focussed on one site the pyritized Hunsrück Slate brittle stars and largely does not consider data from what they term morphological studies to show how our view of the form and function of these ophiuroids has really changed with the CT analysis. There really is no problem with the choice of taxa, the application of the methods which are the standards I expect of the senior author but my concerns are with the Introduction and Discussion. I think the study has been oversold in the introduction and the discussion does not meet my expectations.

I am not convinced that the 3D digital renderings really do improve our understanding of the group. This is a limited survey of just five 'archaic' ophiuroid taxa. The paper claims to have revealed the structure of the arms of these species in 3D for the first time. However we have already known the structure of the arms for over a century (Example Spencer 1914-1920). There are plenty of fossils from classic localities from the Ordovician to the Permian that clearly show 3D preservation of both surfaces. The morphological information has already been collected using traditional methods such as studying latex casts.

Introduction

Line 59: I afraid this is not the case, this might be true for Hunsrück Slate but not for the vast majority of specimens in museum collections. To say that their methods reveals both dorsal and ventral surfaces of each specimen for the first time together with other previously unobserved characters such as the proximal and distal surfaces of the arm segments is not true.

Line 129: These are in effect morphological descriptions that use classification sensu Glass & Blake without mentioning any of the background of the terminology used. In fact, at best these are half way house morphological descriptions and CT scans. It read more like a brief guide to taxa. It does not include background from existing extensive literature on these taxa by Spencer, Blake or Glass. The descriptions have lots of basic errors that demonstrate a general ignorance of the systematics of the group. In addition there is no discussion on the influence of taphonomy on the morphology of these ophiuroids. Recent papers on the fauna of South Africa have not been included.

Discussion

Line 304: The discussion simply repeats observations that have been known for a long time and can be observed in Spencer, Blake or Glass. For we know that “ossicles of many Palaeozoic ‘archaic’ ophiuroids, are paired and offset along the axis of the arm”. It is also well known that the arms are largely incorporated into the body disc. The observation that “there are no extant animals that use two offset columns of repeating segments for musculoskeletally driven locomotion is very interesting but I don’t understand from the MS how this influences movement.

Line 313: It is a very bold claim to say that archaic ophiuroids (Encrinasteridae, Cheiropterasteridae) did not use musculoskeletally driven locomotion similar to that in modern ophiuroids. So and they must have relied on their tube feet for locomotion (‘podial walking’). I don’t see the logic behind this assumption. Why could they have moved their arms? In addition, they have not taken into account that many parts of the ophiuroid skeleton are decalcified. The authors have not taken into account the soft tissue structures that could have existed. It is interesting that the more derived *Furcaster palaeozoicus* among the Palaeozoic ‘archaic’ ophiuroids is the most similar to that in typical extant taxa with bilaterally symmetrical segments, ball-and-socket joints, resembling the structure and location of these joint interfaces and muscle attachment sites. However this is rather a large jump in logic without a functional test of how both these arms functioned. This makes the discussion highly speculative.

Line 326: Why does the morphology of the arm segments of *Ophiurina lymani* provide compelling evidence for podial walking? It is a bit of a jump in logic to say that they are remarkably similar in shape to those of *Amphicutus stygobita*, the extant cave-dwelling brittle star and thus must be a tube feet walker. So many other ophiuroids have this morphology and are not tube foot walkers!

Line 332: The most important result of this study is the evidence of disparity of locomotory function in the pyritized Hunsrück Slate brittle stars. However as warning, details of the articular surface of the vertebrae of living brittle stars do not necessarily correspond to differences in locomotory capability. I think it is true that examining 3D morphology reveals striking contrasts in the morphology of the arms of Encrinasteridae + Cheiropterasteridae, *Furcaster palaeozoicus* and *Ophiurina lymani*, which imply significant differences in function. It is clear there is an evolutionary sequence of development in the arms moving from stepped ambulacrals and inline ambulacrals which is very interesting. But why did you choose the Hunsrück Slate brittle stars when this transformation was very likely to have taken place by the late Silurian? This is in mind it is simply not possible to state that musculoskeletally driven locomotion occurred after the evolution of opposing ambulacrals, but preceded the fusion of the ambulacrals to form vertebrae. or that rows and columns of opposing ambulacrals must have been sufficiently articulated to exhibit this form of motion when you do not have enough data (only 5 taxa) to test this transformation, which might have already taken place by the Devonian.

Conclusions

Line 357: Finally it’s not correct to say “Prior to this study, the 3D morphology of Palaeozoic brittle star arm ossicles was mainly inferred by integrating the morphology of their dorsal and ventral surfaces exposed by preparation of different specimens.” Such information is already available from many other well preserved specimens using traditional methods. I look forward to your updated resubmitted manuscript with more convincing evidence and data.

Reviewer: 3

Comments to the Author(s)

See my in text comments

Author's Response to Decision Letter for (RSOS-200329.R0)

See Appendix B.

RSOS-201380.R0

Review form: Reviewer 2

Is the manuscript scientifically sound in its present form?

No

Are the interpretations and conclusions justified by the results?

No

Is the language acceptable?

Yes

Do you have any ethical concerns with this paper?

No

Have you any concerns about statistical analyses in this paper?

No

Recommendation?

Reject

Comments to the Author(s)

Dear Author

Unfortunately, I have to recommend that this MS is rejected. I believe in its current form this MS meets does not meet the requirements of Royal Society Open Science of a paper that is both scientifically strong and represent a meaningful contribution to the literature. Despite extensive revisions the authors appear unwilling to address the core issues and fundamental flaws in the MS outlined by all 3 reviewers and the Associate Editor. This should have prompted a re-think of what the authors are trying to achieve rather than a major revision that has been presented. I have detailed below the comments which I believe have not been addressed as in order to do so there would need to be a major overhaul of the MS. More worrying there are major areas of the MS were comments from all 3 of the reviewers have simply been "brushed over". Changes made have had little impact on the major issues. For Example:

1. You rephrasing of lines 64-78 do little to solve the issue of Reviewer 3 - Line 60: "But many fossil ophiuroid taxa are known from multiple specimens, each showing different sides and aspects of the animal. Furthermore, preparation can, and has often, been done from both sides on the same animal (even in the Hunsrück Slate, for example in older specimens at the British Museum) I understand that you are trying to make a strong argument for the use of micro scanning here but I DO think you are overstating the case here a little bit when it comes to the "limitations" of other methods."
2. The same is true for lines 75-78: which tried to address the comment of Reviewer 3 - Line 62: I am not sure that these approaches consistently "fail to reveal critical details". They do when the specimen is fully articulated and spacing between ossicles is narrow. There are countless examples however of specimens that are partially disarticulated (including in the Hunsrück Slate)

that reveal these features. I agree with reviewer 3 that it is not true that to view the articulations between successive elements in situ in 3D. This is not possible to do using traditional methods.

3. In addition, I can't see how changes to line (line 75-78) have addressed the concerns of Reviewer 3 - Line 64: that BOTH preparation AND scanning can leave room for error. The authors also use the same lines of text (line 75-78) to answer Reviewer 3 Line 189 concern that observations you claim is novel have already been described by authors such as Roemer, Stuertz, and Lehmann. It appears you have made no effort to clearly separate these novel observations from those that have already been made by others, and/or can be readily observed through study of specimens in x-ray, mechanical, and airbrasive methods.

4. I do not think the comments and changes you have provided address the serious concerns of Reviewer 3 Line. 315: "These are not novel conclusions or interpretations for these taxa and it is not clear how new findings by the CT scanning above was used to derive these conclusions. Please take a look at Harper and Morris (1978) as well as discussions about arm mobility and tube foot walking in Byrne and Hendler (1983)"

5. Reviewer 1 and 2 I cannot see in what way the 3D digital renderings provide new insights. I feel their surplus evidence is not conveyed in a convincing way here. For example, it was clear from previous works that ophiuroids with alternating ambulacrals most probably employed podial walking. You even cite references that drew that conclusion from their own morphological observations, so I cannot see in what way your data provide new insights here. Furthermore, it appears from the authors response down plays the work of Glass and Blake who significantly advanced the work of Spencer and Lehmann. I do not think future researchers need any encouragement to investigate the Hunsrück Slate asterozoans anew as we already have enough knowledge. A shift in methodology will only enhance our existing knowledge. But the existing knowledge and expertise needs to be reviewed and stated by the authors not overlooked.

6. Reviewer 1 points out that such advances have and are being made with reports of dissociated skeletal parts (e.g. Boczarowski 2001) I reject your response that Isolated ossicles are not available for all taxa (they certainly are), and the identification of isolated ossicles in the absence of complete specimens can be difficult but not impossible and these data can be very informative.

7. As pointed out by Reviewer 2 Line 304: The observation that "there are no extant animals that use two offset columns of repeating segments for musculoskeletally driven locomotion is very interesting However I disagree that you can infer that they did not exhibit musculoskeletally-driven motion analogous to that used by extant animals.

8. I can see like Reviewer 1 and Reviewer 2 that there are fundamental problems with conclusion in lines 340 - 342: that is "the expression of musculoskeletally driven locomotion occurred after the evolution of opposing ambulacrals, but preceded the fusion of the ambulacrals to form vertebrae" when you analyse only taxa with ambulacrals either alternating or fused into vertebrae? The key issue is that you have made a broad conclusion based in a limited dataset from just one Lagerstätte the Hunsrück Slate. I do not think you can justify your conclusions by citing Shackleton (2005) as this dataset is Ordovician to Early Silurian or Glass (2005, 2006) which is also restricted to the Hunsrück Slate. In fact, The authors fail to comment on the observation by (Reviewer 2 Line 332). The most important result of this study is the evidence of disparity of locomotory function in the pyritized Hunsrück Slate brittle stars. However, as warning, details of the articular surface of the vertebrae of living brittle stars do not necessarily correspond to differences in locomotory capability. Reviewer 2 points out that there could be a clear there is an evolutionary sequence of development in the arms moving from stepped ambulacrals and inline ambulacrals. However it is almost impossible to identify this trend in the Hunsrück Slate brittle stars due to the limited period of geological time this assemblage covers and the limited dataset (just 5 specimens). Furthermore, such a transformation might have already occurred in the Silurian. I do not see why the "The density of the material provides an appropriate contrast to

conduct an exploratory analysis of the advantages of using micro-CT scanning to examine fossil ophiuroid anatomy or the diversity and disparity of ophiuroid taxa within this locality provide an opportunity for a rich comparative analysis” I reject your conclusion that limiting the investigation to the Hunsrück Slate allows the morphology of each specimen to be directly comparable in terms of their preservation (lines 162-175). Preservation has no influence on the amount of available systematic data. There is amazing brittle star preservation in other localities in the Paleozoic. In order to observe these larger scale trends you need to examine more taxa from more faunas. Its clear that the authors are not listening to the reviewers superior knowledge on any of these points.

9. As per Reviewer 2: Line 357: It’s not correct to say “Prior to this study, the 3D morphology of Palaeozoic brittle star arm ossicles was mainly inferred by integrating the morphology of their dorsal and ventral surfaces exposed by preparation of different specimens.” Such information is already available from many other well-preserved specimens using traditional methods. I don’t any amount of further discussion will hide the fact that all 3 authors do not agree that 3D imaging has revealing previously inaccessible morphological information, such as the articulations between successive segments and the full morphology of a single arm. As we all believe otherwise. This is exemplified by Reviewer 3 Line 361: This is simply not true. The amount of information provided by the CT scanning on the nature, size, shape, and position of the articulation surface between ambulacrals remains equivocal based on the results presented here. Also, there ARE published accounts of what these articulations looked like in specimens from the groups herein, but not necessarily from the Hunsrück Slate, so CT might not be necessary to provide new insights on new interpretations of these. Even your response that “Articulations” is not synonymous with “articular surfaces” This might be correct, but you can still observe Articulations in disarticulated fossil material.

As sadly is the case with many micro-CT scanning studies. They might show spectacular visual results, but these seldom add to anything new. This MS is still highly speculative and urgently needs additional further functional/experimental analyses. I agree with the associate editor’s previous findings that while the case is made in the manuscript that micro-CT visualization of these fossils reveals new details, that many of these morphological interpretations can be made (and have already been made) using more traditional methods. Its clear from your response to reviews that the authors “fundamentally disagree with the notion that your findings are not novel. I agree this is the first application of the only known technique in which ophiuroid arm morphology can be observed in 360°”. However, these techniques might have spectacular the results, but they do not represent a significant advance in what we already known about these taxa. All three of the previously reviewers agree that micro-CT visualization of an ophiuroids although worthwhile the morphology of these taxa has already been observed 360° using latex casts and disarticulated specimens. The authors have then said this MS is the first to observe ophiuroid arm morphology can in 360°. However, these taxa (Genera) do also exist outside the Hunsrück Slate. The data comes from just only a Devonian single deep-water fauna but then tries to make predictions from taxa (Genera) that have existed from the Ordovician to the Permian (and possibly the Triassic). Although I commend the authors for addressing the recommendations of reviewer 2 and 3. The authors have attempted to add more citations to make up for these failings. It still does not include the vast body of literature that exists on these taxa or knowledge of non-Hunsrück Paleozoic ophiuroids from the Ordovician to the Permo-Triassic.

Finally, the apparent ignorance of ophiuroid systematics is demonstrated above and by the following comment “We are not familiar with any evidence that “many parts of the ophiuroid skeleton are decalcified,” and are not aware of any non-calcareous skeletal tissue in extant ophiuroids.” Paleozoic ophiuroids do in fact have areas of the skeleton which widely thought to be decalcified. This sadly, could be attributed to the relative inexperience of the relatively unqualified team which does not include a specialist on ophiuroid systematics, phylogeny and evolution. In fact, only one team member has any knowledge of echinoderms. But this is expertise on the Biomechanics of echinoderms not their morphology and systematics. I seriously question the contribution and usefulness of the other authors in this MS. This would explain the large

number of comments from all 3 of the previous reviewers which demonstrate that the research team does not have the strength and depth I would expect of a specialist of this group. This MS still has many basic mistakes. This MS highlights why such systematic specialists are still useful in paleobiology. These specialists should be integrated into such a research team not over looked or ignored. I recommend that this research team seek the help of such specialist on ophiuroid systematics, phylogeny and evolution when rewriting their MS. The MS would really benefit from someone who has work on taxa from across the Paleozoic. When reading the revised version, I find much of the discussion is highly speculative as the micro CT can only observed the anatomy and does not test the form and function. This was the real achievement of Clark et al. 2018 looking at analytical functional analyses of the morphology. Replicating the methods from this MS would remove all the highly speculative arguments in this paper regarding arm function and mobility and would better reflect the skills and expertise of the first author.

Review form: Reviewer 4 (Frederick Hotchkiss)

Is the manuscript scientifically sound in its present form?

Yes

Are the interpretations and conclusions justified by the results?

Yes

Is the language acceptable?

Yes

Do you have any ethical concerns with this paper?

No

Have you any concerns about statistical analyses in this paper?

No

Recommendation?

Accept with minor revision (please list in comments)

Comments to the Author(s)

File attached. The manuscript is in fine shape. A bit of work is needed regarding Figure 2 and lines 264-268. Hopefully you may find something of value in the suggestions and comments.

For redundancy I paste the review here as well.

Royal Societ review RSOS-201380

Line 36 suggest -- four plesiomorphic

Line 38 suggest -- two separately apomorphic

Line 42 suggest -- This 3D micro-CT scan approach

Line 56 suggest -- Apomorphic living ophiuroids

Line 57 suggest -- apomorphic dorsal, ventral

Line 59 suggest -- change 'Many' to Plesiomorphic Palaeozoic ophiuroids

Line 60 suggest -- delete "...in place of.....ventral plates."

Line 91 suggest -- add ref 16 -- [16-19, 21]

Line 139 comment -- alternating in conformance with Lovén's law -- no action required
 Line 139 suggest -- (alternating, plesiomorphic, Lysophiurina)

Line 140 suggest -- (opposing, apomorphic, Zeugophiurina)

Line 140 -- please reveal if this is supported by you -- or do you think it is controversial

Lines 264-268 -- your arrows in your Fig. 2 do not match up with placement of groove spines in WK Spencer, his clear text-fig. 262, and his verbal assurance text at bottom of page 408. My experience supports WKS placement. Groove is wide open. Please review. Fix as needed.

Lines 279 – 298. Is it within scope to make any comparisons here [or in discussion] with ref 46 -- wherein are observations on the hemi-cylindrical ambulacral ossicles of Hunsrück Loriolaster. ...?

Lines 300 – 311. Is it within scope to make any comparisons here [or in discussion] with ref 16 -- use Eospondylus as proxy for Furcaster . Of note is the arrangement in Furcaster and in Eospondylus of the canal for the radial water vessel being located in the center of the zygosphere knob and the zygotreme pit, like spout and funnel.

Lines 359 - 369 -- I support this interpretation of Ophiurina vis-à-vis Amphicutus. Please reference Südkamp's 2017 book *Leben im Devon/Life in the Devonian*, page 143 where this view is mentioned. The similarity of Ophiurina and Amphicutus is pretty remarkable.

Line 367 -- For myself, the Ophiomusaidae and Ophiosphalmidae seem very similar to each other, and not so similar at all to Ophiurina. You say "with arm morphologies similar to that of *O. lymani*" [as in *Ophiurina lymani* ... yes?]. Is this coming from ref. 60 ...or your analysis?

Line 397 -- add here ----- information comes also from stereo-pair SEM photos of isolated arm ossicles [16, 21]. Stereo-pair images can be created from the 3D micro-CT files as well --- yes?

Line ??? -- 3D micro-CT search for objects of special interest would be of special interest ! Find the madreporite; the madreporite enables ray identification; collect observations related to Lovén's law; find the terminal plates. Keep up the search for a periproct. ...

Review form: Reviewer 5

Is the manuscript scientifically sound in its present form?

No

Are the interpretations and conclusions justified by the results?

Yes

Is the language acceptable?

Yes

Do you have any ethical concerns with this paper?

No

Have you any concerns about statistical analyses in this paper?

No

Recommendation?

Major revision is needed (please make suggestions in comments)

Comments to the Author(s)

Dear Authors,

A number of comments

- 1) Terminology remains unconventional – 1) ‘central axis of the arm’ is the ‘perradial line’ as used by Schuchert and explained clearly in his glossary. 2) Viewed from the ventral side the ambulacrals are subquadrate to subrectangular. Glass is in error here – the boot-shape he attributes to the ambulacrals refers to ridges on the ambulacrals not to the overall shape of the ossicle. 3) ‘Row’ for transverse set of at least 2 Ambb + 2 Adambb is preferable to ‘segment’ which has arthropod overtones. There are others remaining.
- 2) Taphonomic processes. You acknowledge [line241] separation of Ambb columns along the perradial line but but as far as I can see do not acknowledge it anywhere else. Separation of Ambb along the columns in *E. tischbeinianus* is surely taphonomic if you compare with the Sth African specimens of this species Ambb columns figured in Jell & Theron. One reviewer points out that taphonomic processes have affected most Hunsruck specimens and he is correct. Your CT scans show much more dislocation than you have explained. It is unreasonable to suggest that the Ambb in your Fig 2b are ‘in situ’ if that means in life position; the gaps between them are undoubtedly taphonomic. Compare the attitude of the Adambb to the perradial line on either side of the arm in your Fig. 2a and you can clearly see evidence of taphonomic disturbance – this is commonplace across stelleroids preserved in any manner and illustrated in any manner.
- 3) Growth. From the scales given in your Fig. 2a vs 2b the arm of *E. tischbeinianus* (2b) is somewhere near twice the size of the arm of *E. roemeri* (2a). They clearly represent different growth stages because Lehmann 1957 illustrated similarly sized specimens of these two species; they therefore might be expected to be different. For an accurate comparison, adult specimens of the two species ought to be compared.
- 4) I agree with the earlier reviewer who pointed out that your whole discussion of locomotion in these animals is previously understood and published interpretation and as far as I can see that reviewer is correct in asserting that the CT scans have no bearing on that discussion. I suggest you write a review of ophiuroid locomotion without reference to your scans and then when you have finished look to see if the scans add anything extra. That is to say instead of starting with the scanning and assuming it tells you more than was previously known a priori, see what is known then see if the scanning adds anything.
- 5) It appears that the initial interest was to see what CT scanning of these fossilised animals would produce in relation to earlier CT scanning of living ophiuroids and the various questions surrounding Palaeozoic ophiuroids has been investigated superficially as noted by two earlier reviewers who sought more integration with the considerable body of knowledge already available on Palaeozoic ophiuroids and largely ignored herein e.g. Spencer provided excellent illustrations of the articulating proximal and distal faces of Ambb in *Encrinaster grayae* but you fail to mention these or compare with *E. roemeri*. I could go on but the list is quite long.
- 6) Your assertion that *Encrinaster* and *Euzonosoma* are not synonymous in line with Blake and Glass makes the same mistake as they did in canvassing only the two Hunsruck species (Spencer did the same thing in erecting *Euzonosoma* when he compared the two Scottish species). You need to canvas the entire species content and I suggest you will find a full gradation in the length of arms, number of rows per arm, arm width etc. Only after such an examination ought you to make a decision.

Decision letter (RSOS-201380.R0)

Dear Dr Clark

The Editors assigned to your paper RSOS-201380 "Three-dimensional morphology and locomotion of ophiuroids from the Devonian Hunsrück Slate" have now received comments from reviewers and would like you to revise the paper in accordance with the reviewer comments and any comments from the Editors. Please note this decision does not guarantee eventual acceptance.

Please submit your revised manuscript and required files (see below) no later than 21 days from today's (ie 29-Sep-2020) date. Note: the ScholarOne system will 'lock' if submission of the revision is attempted 21 or more days after the deadline. If you do not think you will be able to meet this deadline please contact the editorial office immediately.

on behalf of Dr Jeffrey Thompson (Associate Editor) and Kevin Padian (Subject Editor)
openscience@royalsociety.org

Subject Editor Comments to Author (Professor Kevin Padian):

Thanks for your resubmission and the effort to address the reviewers' comments. On balance I think there is still more work to do and there is some concern about claims that yours is a "better" analysis when perhaps it is not bringing so much that is new. I don't weigh in on this but I hope that you can make its novelty clearer. Thanks and best wishes.

Associate Editor Comments to Author (Dr Jeffrey Thompson):

Dear Elizabeth et al.

At this point, after two rounds of review, the manuscript has been seen by five expert reviewers over two rounds. As it currently stands, 2/5 have recommended rejection, 2/5 have

suggested minor revisions, and a fairly thorough fifth review has suggested major revisions. I am largely happy with the revisions you've made between rounds 1 and 2, though given the major revision and second recommended rejection, I don't feel that I can recommend publication without some more minor changes at this point in time. I like this manuscript, and would like to see it published in Open Science, I am thus recommending minor revisions prior to resubmission. I think that with a few small changes, primarily concerning the framing of the goals of the paper, the manuscript will be acceptable for publication and (most of) the reviewers will be happy. There seems a general feeling amongst the reviewers that the championing of micro-ct scanning in the manuscript implies that previous methodologies are substandard, and ct-scanning has not revealed anything new that couldn't have been learned prior. While I agree with the reviewers that ct-scanning isn't a holy grail capable of solving all the problems of paleobiology, though after reading your responses to the first round of reviews I agree with you that it absolutely facilitates the kinds of hypothesis-driven biomechanical analyses that wouldn't be possible without 3D data. I would suggest that in the next round of edits, you explicitly stress that in order to carry out biomechanical analyses in the future, you need the kinds of data you have here. Perhaps instead of stating that 3d-visualizations shed new light by allowing for 360 visualization of the skeleton (Line 77-78), I might add a sentence or two stating that they allow for biomechanical analyses of animal function (with a few citations across animal groups). I think if this is a little more explicit, then the MS will live up to the expectations promised in the introduction. There also still exists a feeling amongst reviewers that nothing explicitly new regarding the locomotory capacity of Palaeozoic ophiuroids has been gleaned from these analyses. I don't personally believe that all new data need to change interpretations, and if these new data support previous interpretations, I think that doesn't preclude publication. I would thus suggest making it a bit more clear in the discussion and conclusions section when your results and interpretations are in agreement with those of previous authors (e.g. reference # 51).

Additionally, the two new reviews obtained in this round (reviewers #4 and #5) have each requested some explicit changes, and some optional additions which I would suggest you incorporate into the MS before resubmission. In particular, I appreciate reviewer #5 mention that taphonomic artifacts, such as post-mortem disarticulation, aren't discussed much in the manuscript with regard to your interpretations. I also agree with reviewer #5 about the use of "segments" in the manuscript. It's a bit confusing, as I'm not sure if this related to some aspect of the morphology, or to "segmenting out" in making the 3D model. This clarification could help.
Jeff Thompson

Reviewer comments to Author:

Reviewer: 2

Comments to the Author(s)

Dear Author

Unfortunately, I have to recommend that this MS is rejected. I believe in its current form this MS meets does not meet the requirements of Royal Society Open Science of a paper that is both scientifically strong and represent a meaningful contribution to the literature. Despite extensive revisions the authors appear unwilling to address the core issues and fundamental flaws in the MS outlined by all 3 reviewers and the Associate Editor. This should have prompted a re-think of what the authors are trying to achieve rather than a major revision that has been presented. I have detailed below the comments which I believe have not been addressed as in order to do so there would need to be a major overhaul of the MS. More worrying there are major areas of the MS were comments from all 3 of the reviewers have simply been "brushed over". Changes made have had little impact on the major issues. For Example:

1. You rephrasing of lines 64-78 do little to solve the issue of Reviewer 3 - Line 60: "But many fossil ophiuroid taxa are known from multiple specimens, each showing different sides and aspects of the animal. Furthermore, preparation can, and has often, been done from both sides on the same animal (even in the Hunsrück Slate, for example in older specimens at the British Museum) I understand that you are trying to make a strong argument for the use of micro

scanning here but I DO think you are overstating the case here a little bit when it comes to the "limitations" of other methods."

2. The same is true for lines 75-78: which tried to address the comment of Reviewer 3 - Line 62: I am not sure that these approaches consistently "fail to reveal critical details". They do when the specimen is fully articulated and spacing between ossicles is narrow. There are countless examples however of specimens that are partially disarticulated (including in the Hunsrück Slate) that reveal these features. I agree with reviewer 3 that it is not true that to view the articulations between successive elements in situ in 3D. This is not possible to do using traditional methods.

3. In addition, I can't see how changes to line (line 75-78) have addressed the concerns of Reviewer 3 - Line 64: that BOTH preparation AND scanning can leave room for error. The authors also use the same lines of text (line 75-78) to answer Reviewer 3 Line 189 concern that observations you claim is novel have already been described by authors such as Roemer, Stuertz, and Lehmann. It appears you have made no effort to clearly separate these novel observations from those that have already been made by others, and/or can be readily observed through study of specimens in x-ray, mechanical, and airbrasive methods.

4. I do not think the comments and changes you have provided address the serious concerns of Reviewer 3 Line. 315: "These are not novel conclusions or interpretations for these taxa and it is not clear how new findings by the CT scanning above was used to derive these conclusions. Please take a look at Harper and Morris (1978) as well as discussions about arm mobility and tube foot walking in Byrne and Hendler (1983)"

5. Reviewer 1 and 2 I cannot see in what way the 3D digital renderings provide new insights. I feel their surplus evidence is not conveyed in a convincing way here. For example, it was clear from previous works that ophiuroids with alternating ambulacrals most probably employed podial walking. You even cite references that drew that conclusion from their own morphological observations, so I cannot see in what way your data provide new insights here. Furthermore, it appears from the authors response down plays the work of Glass and Blake who significantly advanced the work of Spencer and Lehmann. I do not think future researchers need any encouragement to investigate the Hunsrück Slate asterozoans anew as we already have enough knowledge. A shift in methodology will only enhance our existing knowledge. But the existing knowledge and expertise needs to be reviewed and stated by the authors not overlooked.

6. Reviewer 1 points out that such advances have and are being made with reports of dissociated skeletal parts (e.g. Boczarowski 2001) I reject your response that Isolated ossicles are not available for all taxa (they certainly are), and the identification of isolated ossicles in the absence of complete specimens can be difficult but not impossible and these data can be very informative.

7. As pointed out by Reviewer 2 Line 304: The observation that "there are no extant animals that use two offset columns of repeating segments for musculoskeletally driven locomotion is very interesting However I disagree that you can infer that they did not exhibit musculoskeletally-driven motion analogous to that used by extant animals.

8. I can see like Reviewer 1 and Reviewer 2 that there are fundamental problems with conclusion in lines 340 - 342: that is "the expression of musculoskeletally driven locomotion occurred after the evolution of opposing ambulacrals, but preceded the fusion of the ambulacrals to form vertebrae" when you analyse only taxa with ambulacrals either alternating or fused into vertebrae? The key issue is that you have made a broad conclusion based in a limited dataset from just one Lagerstätte the Hunsrück Slate. I do not think you can justify your conclusions by citing Shackleton (2005) as this dataset is Ordovician to Early Silurian or Glass (2005, 2006) which is also restricted to the Hunsrück Slate. In fact, The authors fail to comment on the observation by (Reviewer 2 Line 332). The most important result of this study is the evidence of disparity of locomotory function in the pyritized Hunsrück Slate brittle stars. However, as warning, details of the articular surface of the vertebrae of living brittle stars do not necessarily correspond to

differences in locomotory capability. Reviewer 2 points out that there could be a clear there is an evolutionary sequence of development in the arms moving from stepped ambulacrals and inline ambulacrals. However it is almost impossible to identify this trend in the Hunsrück Slate brittle stars due to the limited period of geological time this assemblage covers and the limited dataset (just 5 specimens). Furthermore, such a transformation might have already occurred in the Silurian. I do not see why the “The density of the material provides an appropriate contrast to conduct an exploratory analysis of the advantages of using micro-CT scanning to examine fossil ophiuroid anatomy or the diversity and disparity of ophiuroid taxa within this locality provide an opportunity for a rich comparative analysis” I reject your conclusion that limiting the investigation to the Hunsrück Slate allows the morphology of each specimen to be directly comparable in terms of their preservation (lines 162-175). Preservation has no influence on the amount of available systematic data. There is amazing brittle star preservation in other localities in the Paleozoic. In order to observe these larger scale trends you need to examine more taxa from more faunas. Its clear that the authors are not listening to the reviewers superior knowledge on any of these points.

9. As per Reviewer 2: Line 357: It’s not correct to say “Prior to this study, the 3D morphology of Palaeozoic brittle star arm ossicles was mainly inferred by integrating the morphology of their dorsal and ventral surfaces exposed by preparation of different specimens.” Such information is already available from many other well-preserved specimens using traditional methods. I don’t any amount of further discussion will hide the fact that all 3 authors do not agree that 3D imaging has revealing previously inaccessible morphological information, such as the articulations between successive segments and the full morphology of a single arm. As we all believe otherwise. This is exemplified by Reviewer 3 Line 361: This is simply not true. The amount of information provided by the CT scanning on the nature, size, shape, and position of the articulation surface between ambulacrals remains equivocal based on the results presented here. Also, there ARE published accounts of what these articulations looked like in specimens from the groups herein, but not necessarily from the Hunsrück Slate, so CT might not be necessary to provide new insights on new interpretations of these. Even your response that “Articulations” is not synonymous with “articular surfaces” This might be correct, but you can still observe Articulations in disarticulated fossil material.

As sadly is the case with many micro-CT scanning studies. They might show spectacular visual results, but these seldom add to anything new. This MS is still highly speculative and urgently needs additional further functional/experimental analyses. I agree with the associate editor’s previous findings that while the case is made in the manuscript that micro-CT visualization of these fossils reveals new details, that many of these morphological interpretations can be made (and have already been made) using more traditional methods. Its clear from your response to reviews that the authors “fundamentally disagree with the notion that your findings are not novel. I agree this is the first application of the only known technique in which ophiuroid arm morphology can be observed in 360°”. However, these techniques might have spectacular the results, but they do not represent a significant advance in what we already known about these taxa. All three of the previously reviewers agree that micro-CT visualization of an ophiuroids although worthwhile the morphology of these taxa has already been observed 360° using latex casts and disarticulated specimens. The authors have then said this MS is the first to observe ophiuroid arm morphology can in 360°. However, these taxa (Genera) do also exist outside the Hunsrück Slate. The data comes from just only a Devonian single deep-water fauna but then tries to make predictions from taxa (Genera) that have existed from the Ordovician to the Permian (and possibly the Triassic). Although I commend the authors for addressing the recommendations of reviewer 2 and 3. The authors have attempted to add more citations to make up for these failings. It still does not include the vast body of literature that exists on these taxa or knowledge of non-Hunsrück Paleozoic ophiuroids from the Ordovician to the Permo-Triassic.

Finally, the apparent ignorance of ophiuroid systematics is demonstrated above and by the following comment “We are not familiar with any evidence that “many parts of the ophiuroid skeleton are decalcified,” and are not aware of any non-calcareous skeletal tissue in extant

ophiuroids." Paleozoic ophiuroids do in fact have areas of the skeleton which widely thought to be decalcified. This sadly, could be attributed to the relative inexperience of the relatively unqualified team which does not include a specialist on ophiuroid systematics, phylogeny and evolution. In fact, only one team member has any knowledge of echinoderms. But this is expertise on the Biomechanics of echinoderms not their morphology and systematics. I seriously question the contribution and usefulness of the other authors in this MS. This would explain the large number of comments from all 3 of the previous reviewers which demonstrate that the research team does not have the strength and depth I would expect of a specialist of this group. This MS still has many basic mistakes. This MS highlights why such systematic specialists are still useful in paleobiology. These specialists should be integrated into such a research team not over looked or ignored. I recommend that this research team seek the help of such specialist on ophiuroid systematics, phylogeny and evolution when rewriting their MS. The MS would really benefit from someone who has work on taxa from across the Paleozoic. When reading the revised version, I find much of the discussion is highly speculative as the micro CT can only observed the anatomy and does not test the form and function. This was the real achievement of Clark et al. 2018 looking at analytical functional analyses of the morphology. Replicating the methods from this MS would remove all the highly speculative arguments in this paper regarding arm function and mobility and would better reflect the skills are expertise of the first author.

Reviewer: 4

Comments to the Author(s)

File attached. The manuscript is in fine shape. A bit of work is needed regarding Figure 2 and lines 264-268. Hopefully you may find something of value in the suggestions and comments.

For redundancy I paste the review here as well.

Royal Society review RSOS-201380

Line 36 suggest -- four plesiomorphic

Line 38 suggest -- two separately apomorphic

Line 42 suggest -- This 3D micro-CT scan approach

Line 56 suggest -- Apomorphic living ophiuroids

Line 57 suggest -- apomorphic dorsal, ventral

Line 59 suggest -- change 'Many' to Plesiomorphic Palaeozoic ophiuroids

Line 60 suggest -- delete "...in place of.....ventral plates."

Line 91 suggest -- add ref 16 -- [16-19, 21]

Line 139 comment -- alternating in conformance with Lovén's law -- no action required

Line 139 suggest -- (alternating, plesiomorphic, Lysophiurina)

Line 140 suggest -- (opposing, apomorphic, Zeugophiurina)

Line 140 -- please reveal if this is supported by you -- or do you think it is controversial

Lines 264-268 -- your arrows in your Fig. 2 do not match up with placement of groove spines in WK Spencer, his clear text-fig. 262, and his verbal assurance text at bottom of page 408. My experience supports WKS placement. Groove is wide open. Please review. Fix as needed.

Lines 279 - 298. Is it within scope to make any comparisons here [or in discussion] with ref 46 -- wherein are observations on the hemi-cylindrical ambulacral ossicles of Hunsrück Loriolaster. ...?

Lines 300 – 311. Is it within scope to make any comparisons here [or in discussion] with ref 16 -- use *Eospondylus* as proxy for *Furcaster*. Of note is the arrangement in *Furcaster* and in *Eospondylus* of the canal for the radial water vessel being located in the center of the zygosphere knob and the zygotreme pit, like spout and funnel.

Lines 359 - 369 -- I support this interpretation of *Ophiurina* vis-à-vis *Amphicutus*. Please reference Südkamp's 2017 book *Leben im Devon/Life in the Devonian*, page 143 where this view is mentioned. The similarity of *Ophiurina* and *Amphicutus* is pretty remarkable.

Line 367 -- For myself, the *Ophiomusaidae* and *Ophiosphalmidae* seem very similar to each other, and not so similar at all to *Ophiurina*. You say "with arm morphologies similar to that of *O. lymani*" [as in *Ophiurina lymani* ... yes?]. Is this coming from ref. 60 ...or your analysis?

Line 397 -- add here ----- information comes also from stereo-pair SEM photos of isolated arm ossicles [16, 21]. Stereo-pair images can be created from the 3D micro-CT files as well --- yes?

Line ??? -- 3D micro-CT search for objects of special interest would be of special interest ! Find the madreporite; the madreporite enables ray identification; collect observations related to Loven's law; find the terminal plates. Keep up the search for a periproct. ...

Reviewer: 5

Comments to the Author(s)

Dear Authors,

A number of comments

1) Terminology remains unconventional - 1) 'central axis of the arm' is the 'perradial line' as used by Schuchert and explained clearly in his glossary. 2) Viewed from the ventral side the ambulacrals are subquadrate to subrectangular. Glass is in error here - the boot-shape he attributes to the ambulacrals refers to ridges on the ambulacrals not to the overall shape of the ossicle. 3) 'Row' for transverse set of at least 2 *Ambb* + 2 *Adambb* is preferable to 'segment' which has arthropod overtones. There are others remaining.

2) Taphonomic processes. You acknowledge [line241] separation of *Ambb* columns along the perradial line but but as far as I can see do not acknowledge it anywhere else. Separation of *Ambb* along the columns in *E. tischbeinianus* is surely taphonomic if you compare with the Sth African specimens of this species *Ambb* columns figured in Jell & Theron. One reviewer points out that taphonomic processes have affected most Hunsruck specimens and he is correct. Your CT scans show much more dislocation than you have explained. It is unreasonable to suggest that the *Ambb* in your Fig 2b are 'in situ' if that means in life position; the gaps between them are undoubtedly taphonomic. Compare the attitude of the *Adambb* to the perradial line on either side of the arm in your Fig. 2a and you can clearly see evidence of taphonomic disturbance - this is commonplace across stelleroids preserved in any manner and illustrated in any manner.

3) Growth. From the scales given in your Fig. 2a vs 2b the arm of *E. tischbeinianus* (2b) is somewhere near twice the size of the arm of *E. roemeri* (2a). They clearly represent different growth stages because Lehmann 1957 illustrated similarly sized specimens of these two species; they therefore might be expected to be different. For an accurate comparison, adult specimens of the two species ought to be compared.

4) I agree with the earlier reviewer who pointed out that your whole discussion of locomotion in these animals is previously understood and published interpretation and as far as I can see that reviewer is correct in asserting that the CT scans have no bearing on that discussion. I suggest you write a review of ophiuroid locomotion without reference to your scans and then when you have finished look to see if the scans add anything extra. That is to say instead of starting with

the scanning and assuming it tells you more than was previously known a priori, see what is known then see if the scanning adds anything.

5) It appears that the initial interest was to see what CT scanning of these fossilised animals would produce in relation to earlier CT scanning of living ophiuroids and the various questions surrounding Palaeozoic ophiuroids has been investigated superficially as noted by two earlier reviewers who sought more integration with the considerable body of knowledge already available on Palaeozoic ophiuroids and largely ignored herein e.g. Spencer provided excellent illustrations of the articulating proximal and distal faces of *Ambb* in *Encrinaster grayae* but you fail to mention these or compare with *E. roemeri*. I could go on but the list is quite long.

6) Your assertion that *Encrinaster* and *Euzonosoma* are not synonymous in line with Blake and Glass makes the same mistake as they did in canvassing only the two Hunsruck species (Spencer did the same thing in erecting *Euzonosoma* when he compared the two Scottish species). You need to canvas the entire species content and I suggest you will find a full gradation in the length of arms, number of rows per arm, arm width etc. Only after such an examination ought you to make a decision.

===PREPARING YOUR MANUSCRIPT===

===PREPARING YOUR REVISION IN SCHOLARONE===

Author's Response to Decision Letter for (RSOS-201380.R0)

See Appendix C.

Decision letter (RSOS-201380.R1)

Dear Dr Clark,

It is a pleasure to accept your manuscript entitled "Three-dimensional visualization as a tool for interpreting locomotion strategies in ophiuroids from the Devonian Hunsrück Slate" in its current form for publication in Royal Society Open Science.

on behalf of Mr Jeffrey Thompson (Associate Editor) and Kevin Padian (Subject Editor)
openscience@royalsociety.org

Associate Editor Comments to Author (Mr Jeffrey Thompson):

Dear Elizabeth et al.
Following your revisions, I now feel that this manuscript is ready to be published in open science. I appreciate your work in addressing the comments of reviewers and editors.
Jeff Thompson

Appendix A**ROYAL SOCIETY
OPEN SCIENCE****Three-dimensional morphology and locomotion of
ophiuroids from the Devonian Hunsrück Slate**

Journal:	Royal Society Open Science
Manuscript ID	RSOS-200329
Article Type:	Research
Date Submitted by the Author:	27-Feb-2020
Complete List of Authors:	Clark, Elizabeth; Yale University, Geology and Geophysics Hutchinson, John; The Royal Veterinary College, Comparative Biomedical Sciences Briggs, Derek; Yale University, Geology and Geophysics
Subject:	Palaeontology < EARTH SCIENCES, biomechanics < PHYSICS
Keywords:	Ophiuroidea, 'archaic' ophiuroids, locomotion, 3D imaging
Subject Category:	Organismal and Evolutionary Biology

Author-supplied statements

Relevant information will appear here if provided.

Ethics

Does your article include research that required ethical approval or permits?:

This article does not present research with ethical considerations

Statement (if applicable):

CUST_IF_YES_ETHICS :No data available.

Data

It is a condition of publication that data, code and materials supporting your paper are made publicly available. Does your paper present new data?:

Yes

Statement (if applicable):

Data accessibility. The micro-CT scans are available from FigShare:

<https://figshare.com/s/ad6f1a971cece0a8827b>

Conflict of interest

I/We declare we have no competing interests

Statement (if applicable):

CUST_STATE_CONFLICT :No data available.

Authors' contributions

This paper has multiple authors and our individual contributions were as below

Statement (if applicable):

Authors' contributions. EGC, JRH, and DEGB were involved in conceptualization and funding acquisition. EGC processed the micro-CT data. EGC and JRH constructed the digital models. EGC, JRH and DEGB interpreted the results and prepared the manuscript.

**Title: Three-dimensional morphology and locomotion of ophiuroids from the Devonian**
**Hunsrück Slate**

**Authors:** E. G. Clark¹, J. R. Hutchinson², D. E. G. Briggs^{1,3}

¹Department of Geology and Geophysics, Yale University, 210 Whitney Avenue, New Haven,
CT 06511, USA

²Structure and Motion Laboratory, Department of Comparative Biomedical Sciences, The
Royal Veterinary College, Hawkshead Lane, Hatfield, Hertfordshire, AL9 7TA, UK

³Yale Peabody Museum of Natural History, Yale University, New Haven, CT 06511, USA

Contact information of corresponding author:

Elizabeth G. Clark, 210 Whitney Avenue, New Haven, CT 06511

Telephone: 412-956-2263

Email: elizabeth.g.clark@yale.edu

**Keywords:** Ophiuroidea, 'archaic' ophiuroids, locomotion, 3D imaging

**Abstract**

Living brittle stars (Echinodermata: Ophiuroidea) employ a very different locomotion strategy to
that of any other metazoan: five or more arms coordinate powerful strides for rapid movement
across the ocean floor. This mode of locomotion is reliant on the unique morphology of
multifaceted skeletal structures and associated muscles and other soft tissues. The skeleton of
many Palaeozoic ophiuroids differs markedly from that in living forms, making it difficult to infer
their mode of locomotion and, therefore, to resolve the evolutionary history of locomotion in the
group. Here, we present 3D digital renderings of five 'archaic' ophiuroid taxa, *Encrinaster roemeri*,
*Euzonosoma tischbeinianum*, *Loriolaster mirabilis*, *Cheiropteraster giganteus*, *Furcaster*
*palaeozoicus*, together with *Ophiurina lymani*, all from the Lower Devonian Hunsrück Slate,
revealing the structure of the arms of these species in 3D for the first time. Their morphological
disparity indicates that there were at least three distinct mechanisms for locomotion among
Hunsrück Slate ophiuroids. This approach promises new insights into the phylogeny and
ecological role of Palaeozoic brittle stars.

**Introduction**

*Investigating ophiuroid locomotion*

Locomotion strategy reflects the relationship of an organism to its environment (e.g. how
it feeds, reproduces, protects itself from predators, and where it lives). The integrated body
structure of an organism and the morphology of its components determine the locomotion
strategies available to it. The majority of living ophiuroids employ a unique strategy involving the
coordination of whip-like motions of their five muscular arms, in contrast to their closest relatives,
including the sea stars (Asteroidea) [1-3], which primarily use tube feet and move more slowly.

This suggests that modern ophiuroids evolved from tube-foot walking ancestors 4
Reconstructing the evolutionary history of ophiuroid locomotion is challenging as the construction
of the arms of many fossil ophiuroids differs from that in living forms, from the basic design and
number of elements to their arrangement and integration. Hunter and McNamara [4] pointed out
the absence of “convincing arguments to link flexibility to the style or success of ... locomotory
or food-gathering movements” in Palaeozoic ophiuroids. Determining the movement capabilities
and inferring locomotion strategies of extinct ophiuroid morphologies is necessary to reconstruct
the evolutionary history of modern ophiuroid locomotion, and to better understand their ecology.
Specimens of articulated fossil brittle stars are typically prepared mechanically or with airbrasive
tools and only the dorsal or ventral surface is exposed (figure 1). Interpretation of their overall
morphology relies on combining observations of the dorsal and ventral surfaces of separate
specimens. This approach fails to reveal critical details of the proximal and distal surfaces of the
arm segments and the nature of the articulations between them, and leaves room for error in
interpreting the alignment of dorsal and ventral features  We used 3D imaging to investigate
the morphology of the arms of six Palaeozoic brittle star taxa. This method reveals both dorsal and
ventral surfaces of each specimen for the first time together with other previously unobserved
characters such as the proximal and distal surfaces of the arm segments.

*Phylogeny and classification of total group ophiuroids*

Four orders of total group ophiuroids were recognized traditionally  Oegophiuroida [7],
Phrynophiurida [7], Stenurida [8], and Ophiurida [9]. Oegophiuroida was thought to include one
extant taxon, *Ophiocanops* [6,10] but this genus has been reinterpreted as falling within Ophiurida,
family Ophiomyxidae [11-13]. Phrynophiurida is post-Palaeozoic [14] and originally included

Euryalina and Ophiomyxidae [10]. However, the euryalids were found to be a monophyletic clade
nested within the Ophiurida and the Ophiomyxidae were resolved elsewhere within the Ophiurida
based on a molecular phylogenetic analysis [15]. Now asterozoans are typically assigned to four
groups: Somasteroidea, Asteroidea (fossil and living sea stars), Ophiuroidea (fossil and living
brittle stars), and Stenuroidea [16]. Oegophiuroidea and Euryalida are included within Ophiuroidea
[13]. The Stenuroidea and Oegophiuroidea are exclusive to the Palaeozoic [11,13] whereas
euryalids are post-Palaeozoic [13,15-16]. All four asterozoan groups first appear in the Ordovician
[17], but molecular clock estimates suggest that the asterozoans originated in the Cambrian [15]
although no Cambrian asterozoans have been found [17].

Ophiuroids are hypothesized to have evolved from a somasteroid ancestor [16,18]. The
relationship between Stenuroidea and crown ophiuroids remains uncertain [17]. Stenuroids have
been interpreted as stem-group ophiuroids that exemplify an ‘archaic’ morphology, which includes
most notably the presence of unfused ambulacral ossicles [4]. Smith et al. [19] combined
molecular and morphological evidence to generate a phylogeny of ophiuroids incorporating two
fossil taxa: one Palaeozoic, *Strataster ohioensis*, which was used as the outgroup, and one post-
Palaeozoic, family Aplocomidae. Shackleton [18] provided the most comprehensive phylogeny of
Palaeozoic Asterozoa, illuminating relationships between several major families. Blake and
Guensburg [17] presented a hypothesis of relationships between the Somasteroidea, Stenuroidea,
Ophiuroidea and Asteroidea: the somasteroid *Archegonaster* was found to be nested within the
stenuroids, raising questions regarding the monophyly of these group. Although phylogenomic
tools have been applied to extant taxa [13,15], there has been no comprehensive phylogenetic
analysis of total group ophiuroids.

Uncertainty about ophiuroid relationships is due, in part, to the elimination of much
morphological diversity during the P/T extinction [20,21]. Molecular clock estimates suggest that
crown-group ophiuroids originated in the Permian [15]. All Permian families of Ophiurida, except
Aganasteridae, have Triassic representatives [20]. Diversity at the genus/species level was greater
in the lower Triassic than in the Permian, indicating that post-extinction recovery of ophiuroids
was relatively rapid [20]. This was accompanied by the radiation of the crown group, characterized
by the ‘modern’ arm morphology (with fused ambulacrals forming vertebrae) [4]. Thus, a
transition is evident in the fossil record between the ‘archaic’ arm morphology, which was
dominant in the Palaeozoic, and the ‘modern’ arm morphology that was present from the Devonian
onward and diversified rapidly following the P/T extinction [22]. It is not known how the crown
is related to the stem, nor is it clear whether or not the ‘modern’ morphology is an apomorphy of
the crown [22], as has been suggested [4,19]. The functional consequences of the ‘archaic’ arm
morphology have not been investigated, leaving the origin of musculoskeletally driven brittle star
locomotion unknown.

*Arm morphology and movement*

Living ophiuroids typically have five arms made of ~100 repeating segments of five ossicles each
[23]. One large ossicle (the vertebra, formed from fused ambulacrals [22]) forms the center of each
segment. Living ophiuroids generally coordinate periodic musculoskeletally driven oscillations of
their arms to effect rapid locomotion [24-26]; *Amphicutis stygobita*, which uses tube foot walking
[27], represents an exception. Related taxa such as sea stars and sea urchins [2-3,15,28], in contrast,
use tube feet as a primary tool for locomotion. This suggests that musculoskeletally driven

locomotion in ophiuroids is derived from a tube foot-walking ancestor. Elucidating the functional
capabilities of these ‘archaic’ ophiuroid arms is key to illuminating this mechanical transition.

‘Archaic’ Palaeozoic ophiuroids [4] have two rows of ambulacral ossicles along the central
axis of the arm (figures 1,2). The rows of ambulacrals are typically offset (alternating). Certain
taxa have ambulacrals symmetrical across the midline of the arm (opposing); phylogenetic
evidence suggests that this state originated multiple times [18]. The morphology of the ambulacrals
varies, although in many taxa the ventral surface is boot-shaped [2]. Inferences regarding the
movement capabilities and locomotion strategies of ‘archaic’ taxa have been based on the
morphology of their arms [4,6,29]. Spencer and Wright [6], for example, inferred that the
articulations and muscles in Stenurida only allowed “simple movements” and that Oegophiuroida
did “not move speedily on the seafloor” due to constraints imposed by their arm morphology.
Glass and Blake [29] suggested that Palaeozoic ophiuroids had “limited arm mobility” and most
likely used tube foot-walking as do modern asterozoans. Hunter and McNamara [4] suggested that
‘modern’ ophiuroids have greater mobility than ‘archaic’ ones, allowing them to radiate into a
wider variety of ecological niches. However, these interpretations of function were based purely
on morphological descriptions, and range-of-motion analyses of modern ophiuroid arms suggest
that functional disparity cannot be determined from morphological observations alone [3]. In
addition, arm flexibility does not correlate with trophic mode [31]. Further analyses are necessary
to determine the functional capabilities of ‘archaic’ ophiuroid arm morphology and reconstruct the
history of locomotion within the group. The first step towards determining arm function in
Palaeozoic ophiuroids is to understand their morphology in 3D [30]. We approached this by
developing 3D digital reconstructions based on micro-CT images of representative exceptionally
preserved fossil specimens.

**Material and methods**

The Lower Devonian Hunsrück Slate of western Germany has yielded a remarkable
diversity of pyritized fossils during roof slate mining [32,33]. The Hunsrück Slate is one of the
most important sources of completely preserved Palaeozoic ophiuroids [6,34] and it has yielded
examples of soft tissue preservation in ophiuroids, including tube feet [29], as well as ophiuroid
trackways [3]. Traditional preparation of Hunsrück Slate material involves scraping away the
slate matrix with wire brushes and steel picks (figure 1*d*), which runs the risk of removing surface
detail from the specimens. Careful airbrasive preparation with iron powder retains spectacular
detail [29], but is very time-consuming. 3D imaging allows the slate matrix to be removed digitally
from the pyritized specimen; the micro-CT imaging technique translates the large density
differential between the slate matrix and the pyritized specimen into excellent digital contrast
(figure 1). This technique also illuminates the morphology in 3D and allows for easy data
dissemination.

This investigation was based on specimens held by the Goldfuß Museum of the Steinmann
Institute of the University of Bonn. Specimens were selected for micro-CT scanning based on
completeness and the quality of preparation; those that exceeded the size limitations of the scanner
were not considered. Thirty-three specimens found to be suitable were x-rayed (e.g. figure 1*b,e*)
to evaluate the degree of detail preserved; many of these articulated specimens were too lightly
pyritized or overgrown with pyrite internally, including a number that appeared pristine on the
surface. Diagenetic factors leading to merging, compaction, or shearing of skeletal elements
eliminated several others. In this way we determined the best candidates for micro-CT scanning,
including a diversity of brittle stars in which the morphology of individual segments and ossicles

could be observed in the x-ray image.

We acquired digital images of one arm of each of six specimens representing six species
in four families. Encrinasteridae: *Encrinaster roemeri* (Hubo 116), *Euzonosoma tischbeinianum*
(EGR 27) [18,36]; Cheiropterasteridae: *Loriolaster mirabilis* (ESCH 596), *Cheiropteraster*
*giganteus* (Hubo 119) [37]; Protasteridae: *Furcaster palaeozoicus* (OKL 96) [18,36], and
Ophiurinae: *Ophiurina lymani* (HS 705) [6] (figure 2).

The morphology of Hunsrück Slate specimens is partially obscured by the slate matrix. CT
scanning allows 360° visualization and reveals the area between adjacent arm segments [30].
Relevant skeletal structures, which are often sub-mm in size, can be enlarged for examination.
Skeletal features were extracted from the scans and imported for visualization in Maya (Autodesk).
3D imaging and digital visualization was used previously to reveal the morphology of the water
vascular system of the Ordovician ophiuroid *Protasterina* from Kentucky [5] and to interpret
accumulations of ophiuroids in the Lower Devonian Bokkeveld Group of South Africa [38].

The specimens were imaged using the GE phoenix|x-ray v|tome|x micro-CT scanner at the
Steinmann Institute, University of Bonn and visualized using VG Studio MAX in the Briggs and
Bhullar labs at Yale University. Ossicles were extracted from the CT scan as watertight polygon
meshes using VG Studio 3.0. Meshes of individual segments were cleaned in Meshlab [3].

**Results of CT scanning**

*Encrinaster roemeri* [Encrinasteridae] (Hubo 116) (figure 2a)

The arm of *Encrinaster roemeri* comprises two major components: ambulacrals and
adambulacrals. These ossicles are aligned within a plane so that the arm appears relatively flat
when visualized digitally. There are no dorsal or ventral ossicles, in contrast to the arms of the

majority of extant ophiuroids [40]. Nearly half of the length of the arm is incorporated into the
disc.

The ambulacrals form two rows offset along the long axis of the arm. The ventral surface
is boot-shaped in outline: the length of the “leg” and the width of the “foot” are approximately
equal, and the length of the “toe” is approximately half the length of the leg (terminology in [29]
Fig. 6). The width of the “distal fitting” and “center leg” are approximately equal to the length of
the toe. The dorsal surface is trapezoidal, the lateral margin shorter than the central margin. There
are slight mediolateral ridges towards the proximal and distal ends of the dorsal surface. The
proximal and distal surfaces are relatively flat with several ridges which may represent sites of
muscle attachment. The axial face is relatively flat. The “lace area” [29] accommodates a pore-
like structure which presumably represents a podial basin hosting a tube foot as in the Ordovician
ophiuroid *Protasterina flexuosa* [5]. There are spaces between successive ambulacrals within the
same row which can be visualized in the micro-CT scan but are obscured in the fossil. There
appears to be a gap along the mid-line of the arm between the rows of ambulacrals but the digitized
specimen shows that the ossicles fit together tightly: there is little or no space between the rows.
Marked changes in the ambulacrals along the length of the arm include a reduction in the width of
the distal fitting and the size of the podial basin.

Each ambulacral ossicle has a corresponding adambulacral ossicle on the margin of the
arm. These adambulacrals are relatively simple curved structures, slightly convex abradially. The
long axis of the adambulacrals is positioned at approximately 40° to the axis of the arm. The
adambulacrals of the free arm are larger than those within the disk. Pyritized elements extend
between the ventral margin of the adambulacrals and the toe of the ambulacral.

*Euzonosoma tischbeinianum* [Encrinasteridae] (EGR 27) (figures 1a-c, 2b)

The ambulacrals and adambulacrals are arranged in two rows offset across the arm axis, as
in *Encrinaster roemeri*. Ventral and dorsal ossicles are absent. More than half of the length of the
arm projects beyond the disk. The degree of separation of the two rows varies between arms,
presumably as a result of taphonomic processes.

The ventral surface of the ambulacrals is boot-shaped in outline. The length of the leg is
shorter than the width of the foot along most of the arm, but they approach the same length toward
the distal tip. The length of the toe is shorter than the width of the distal fitting and less than half
the width of the foot. The width of the distal fitting and center leg are approximately equal. The
dorsal surface of the ambulacrals is roughly rectangular in outline. Flared proximal and distal
ridges are associated with a depression along the mediolateral axis of the dorsal face of the
proximal ossicles; the dorsal surface becomes gradually smoother in more distal segments. The
proximal and distal faces are relatively flat, with ridges resembling those in *E. roemeri*. The axial
face of the ambulacrals is relatively flat, with a curved depression in the center. The most striking
differences between *E. tischbeinianum* and *E. roemeri* are the greater dorsoventral thickness of the
ambulacrals in *E. tischbeinianum* and the reduction in relative size of the ambulacrals and
adambulacrals toward the distal end of the arm (figure 2a,b). There are spaces between successive
ambulacrals in *E. tischbeinianum* as in *E. roemeri* but, in contrast to *E. roemeri*, there are large
gaps between the two rows of ambulacrals in this specimen.

The adambulacrals of *Euzonosoma tischbeinianum* are relatively flat on both proximal and
distal surfaces. The dorsal, ventral, and lateral surfaces are concave. The adaxial margin is V-
shaped, the distal arm of the “V” slightly longer than the proximal. The adambulacrals become

smaller along the free arm distally. They are wider than long within the free arm, in contrast to
those in the fixed arm.

Previous descriptions based on surface preparations illustrate the adaxial margin of the
adambulacrals directly abutting or even overlapping the ventral part of the ambulacrals in
*Euzonosoma tischbeinianum* and *Encrinaster roemeri* (e.g. [41], p. 422). However, we observed
pyritized needle-like structures between the ambulacrals and the adambulacrals in Hubo 116 and
EGR 27 (figure 2a,b). These pyritized elements vary in morphology in each arm segment as well
as among the other arms that we x-rayed; they represent partially pyritized groove  nes (sensu
[41], p. 408).

Jell and Theron [42, p. 165] synonymized *Euzonosoma* with *Encrinaster*. They noted that
Spencer [41] distinguished these genera on the basis of a single character, the mediolateral width
of the adambulacrals. In *Euzonosoma*, there is a distinct difference between the width of these
ossicles at the tip and at the base of the free arm (figure 2b). In *Encrinaster*, in contrast, the width
is similar along the length of the free arm (figure 2a). Jell and Theron [42] argued that this feature
reflects the angle at which these ossicles are preserved and exposed in the specimens. In the 3D
image of the specimens we investigated, however, it is apparent that the relative size of the
adambulacrals within the free arm does indeed differ between the two taxa in line with the original
designation [41]. Thus we retain the genus *Euzonosoma* here.
*Loriolaster mirabilis* [Cheiropterasteridae] (ESCH 596) (figure 2c)

The ambulacrals are offset. The ventral surface is exposed in the specimen, and the outline of the
ventral ambulacrals is not sharp in the digital image presumably due to overpreparation.
Ambulacrals are rectangular to trapezoidal in dorsal view. The ambulacrals become narrower
along the arm distally. Ambulacrals form an archway along the proximodistal axis where they
abut, presumably housing the radial canal of the water vascular system. The proximal and distal
faces of the ambulacrals appear to be relatively flat. This specimen is unusual in having only four
arms but the morphology of the elements is typical of the taxon. There is no evidence that a fifth
arm was in the process of regeneration.

*Cheiropteraster giganteus* [Cheiropterasteridae] (Hubo 119) (figure 2d)

The ambulacrals have a straight proximal ridge and u-shaped distal ridge in dorsal view. The width
of the ambulacrals tapers abaxially. The distal face has a central depression whereas the proximal
face is relatively flat, angled at approximately 30° to the mediolateral axis. A central depression in
the axial surface creates a slightly sinusoidal space for the radial canal between the paired
ambulacrals. The morphology of individual ossicles on the ventral surface is difficult to discern in
the 3D image, as is part of the area between segments. The adambulacrals appear to be relatively
small and slightly curved and elongate; long spines are preserved projecting from them in many
cases.

*Furcaster palaeozoicus* [Protasteridae] (OKL 96) (figure 2e)
The paired ambulacrals are opposing and roughly rectangular in ventral view (longest along the
proximodistal axis of the arm) with ridges on the proximal and distal faces. The pairs are narrowest
at their mid-length and widest towards the proximal end. The pairs are triangular in dorsal outline
in the proximal part of the arm, wide distally and tapering towards the proximal end; they are
approximately rectangular in the distal part. The distal face is relatively flat with a median

projection similar to that in modern ophiuroids (see [30], Figs 2, 6-8), which has been interpreted
as a ball-and-socket joint in *Furcaster trepidans* [18]. The proximal and distal ridges surrounding
the projection form an area on the proximal and distal faces which resemble the four muscle
attachment sites in modern ophiuroids (see [30], Fig. 2). The adambulacrals are curved, often with
a ventral projection on the proximal abaxial end. The segments reduce in width along the length
of the arm.
*Ophiurina lymani* [Ophiurinae] (HS 705) (figures 1d-f, 2f)
Although the micro-CT scan does not reveal details of individual plates, the extracted 3D-rendered
images show the morphology of the segments. Segments, which are many fewer than those in the
arms of typical ‘archaic’ ophiuroids, are roughly rectangular in both dorsal and ventral view. A
median groove runs along the proximodistal axis on both the dorsal and ventral surfaces. The distal
face appears slightly convex and abuts with the concave face of the next segment. Segments widen
distally (excluding the most distal segment). The proximal segments are wider than those at the tip
of the arm.

36 37 38 39 293 40 41 294 **Discussion**

42 43 295 44 45 296 *Functional Morphology of the ‘Archaic’ Ophiuroid Arm*

Analysing function in fossil animals is typically conducted through adapting and applying
techniques used to study function in living forms (e.g. [43]). However, the structure of the arms of
‘archaic’ ophiuroids is fundamentally different from that of living taxa [4], which presents a
challenge as there are no clear modern analogs. Extant ophiuroid arms are comprised of modular

segments, each with a single central vertebral ossicle, which is bilaterally symmetrical along the
axis of the arm. These vertebrae are connected by ball-and-socket articulations which are
surrounded by muscles. Contraction of the muscles flexes the distal segment about the joint [30].
The ossicles of many Palaeozoic ‘archaic’ ophiuroids, in contrast, are paired and offset along the
axis of the arm (e.g. those of *Encrinaster roemeri*, *Euzonosoma tischbeinianum*, *Loriolaster*
*mirabilis*, and *Cheiropteraster giganteus*) (figure 2a-c). In addition, they lack distinct joint
interfaces, which are integral to performing movement in living forms. The ossicles in the two
central rows in these ophiuroids tend to be separated proximodistally and, in some cases, tightly
articulated across the midline of the arm (in *Encrinaster roemeri*, *Loriolaster mirabilis*).
Furthermore, the arms are largely incorporated into the body disc. Spencer and Wright regarded
these forms as largely sessile bottom dwellers. To our knowledge, there are no extant animals that
use two offset columns of repeating segments for musculoskeletally driven locomotion. The
morphological evidence suggests that these archaic ophiuroids (Encrinasteridae,
Cheiropterasteridae) did not use musculoskeletally driven locomotion similar to that in modern
ophiuroids and they must have relied on their tube feet for locomotion (‘podial walking’).

The arrangement in *Furcaster palaeozoicus* (figure 2e), among the Palaeozoic ‘archaic’
ophiuroids examined here, is the most similar to that in typical extant taxa. The segments are
bilaterally symmetrical. There is evidence in some segments of ball-and-socket joints, resembling
the structure and location of these joint interfaces and muscle attachment sites in extant ophiuroids,
suggesting that musculoskeletally driven locomotion was possible. The length of the free versus
the fixed arm is much greater in *F. palaeozoicus* than in the other taxa examined here; the long,
flexible arms (e.g. [33], fig. 81) unencumbered by disc tissue is consistent with musculoskeletally
driven motion.

Spencer and Wright [6] considered Devonian Ophiurida (designated ‘modern’ Palaeozoic
ophiuroids by [4]) to be capable of active movement due to their flexible arms. The morphology
of the arm segments of *Ophiurina lymani* (Ophiurida) (figures 1d-f, 2f), however, provides
compelling evidence for podial walking. The segments are remarkably similar in shape to those of
*Amphicutus stygobita* ([27] Figs. 3,6,7,10), an extant cave-dwelling brittle star that mainly
locomotes by tube foot walking. The segments in both are bilaterally symmetrical and elongate,
widening distally. Distal widening in *O. lymani* accommodates structures that we interpret as
podial basins (figure 2f), in the same position as those in *A. stygobita*.

Our analysis of 3D morphology of pyritized Hunsrück Slate brittle stars reveals evidence
of a disparity of locomotory function. Details of the articular surface of the vertebrae of living
brittle stars do not necessarily correspond to differences in locomotory capability [30]. However,
our analysis of 3D morphology reveals striking contrasts in the morphology of the arms of
Encrinasteridae + Cheiropterasteridae, *Furcaster palaeozoicus* and *Ophiurina lymani*, which
imply significant differences in function.

As vertebral ossicles are essential for musculoskeletally driven locomotion in extant
ophiuroids, opposition and fusion of the ambulacrals represent critical steps towards the
development of this type of movement. Our new evidence suggests that the expression of
musculoskeletally driven locomotion occurred after the evolution of opposing ambulacrals, but
preceded the fusion of the ambulacrals to form vertebrae flows and columns of opposing
ambulacrals must have been sufficiently articulated to exhibit this form of motion. The application
of the methodology outlined here to a broad scope of Palaeozoic ophiuroids with opposing and
fused ambulacrals will reveal the evolutionary history and ecological context of the rise to
dominance of musculoskeletally driven locomotion in brittle stars.

The application of methodologies to test functional capacity using 3D image data (i.e.[30])
will be important for extending this result to these and other appropriately preserved taxa. In
addition, although the stereom structure of the ambulacrals and adambulacrals is not preserved in
Hunsrück Slate specimens, evidence from other localities may be key to identifying the placement
of connective tissue and muscle (e.g. [44]) present in ‘archaic’ ophiuroid arms. Integration of these
morphofunctional data with emerging insight into the biomechanics of locomotion of echinoderms
at the whole-organism level and on different surfaces (e.g. [3,24-26,45-47]) could provide
transformative understanding into the major evolutionary transitions of echinoderm locomotion.

**Conclusions**

Prior to this study, the 3D morphology of Palaeozoic brittle star arm ossicles was mainly inferred
by integrating the morphology of their dorsal and ventral surfaces exposed by preparation of
different specimens, with some additional information from x-radiographs [32]. Our investigation
used a non-destructive approach to reveal the full morphology of the arm ossicles and their
articulations in 3D, representing the most complete visualization of their morphology to date.  We
showed that the strategy used for locomotion by ‘archaic’ ophiuroids with alternating ambulacrals
is fundamentally distinct from the musculoskeletally driven motion of modern ophiuroids, as they
lack critical structures necessary to produce this form of movement. The paired ambulacrals and
ball-and-socket joints in *Furcaster palaeozoicus*, together with their long free arms, show that
musculoskeletally driven motion was likely possible. The arm segments of the ophiurid *Ophiurina*
*lymani* are also bilaterally symmetrical, but morphological similarities with the extant *Amphicutis*
*stygobita*, including short arms and the position of podial basins, favors tube-foot (podial) walking.
These disparate morphologies indicate fundamental differences in the underlying mechanics of

arm movement. Fusion of opposed vertebrae represents a critical transitional step preceding the
origin of modern brittle star musculoskeletally driven locomotion. Future study of Palaeozoic
brittle stars integrating 3D imaging with dynamic modeling should reveal further insight into the
evolution of locomotion in ophiuroids.
**Figure 1.** (a-c) *Euzonosoma tischbeinianum* (EGR 27), (d-f) *Ophiurina lymani* (HS 705). (a,d)
Photograph. Note that EGR 27 has undergone minimal preparation. (b,e) X-ray image. (c,f) Digital
rendering of the 3D micro-CT image in Autodesk Maya, (c) shows only one arm of EGR 27.

**Figure 2.** Arms of (a) *Encrinaster roemeri* (Hubo 116), (b) *Euzonosoma*
*tischbeinianum* (EGR 27), (c) *Loriolaster mirabilis* (ESCH 596), (d) *Cheiropteraster*
*giganteus* (Hubo 119), (e) *Furcaster palaeozoicus* (OKL 96), and (f) *Ophiurina lymani* (HS 705).
Views of ventral (left) and dorsal (right) surface of arm and details of segments on ventral (above)
and dorsal (below) sides (dorsal and ventral details of *Furcaster palaeozoicus* and dorsal details
of *Loriolaster mirabilis* are from a different arm of the same specimen). Arrows indicate
depressions corresponding to podial basins in *Ophiurina lymani*.

**Acknowledgements.** We are grateful to Alex Glass (Duke University) Fred Hotchkiss (Marine
and Paleobiological Research Institute), Christoph Bartels, Gabriele Kühl, Jes Rust, Teresa Franke,
Alexandra Bergmann, Georg Oleschinski (University of Bonn), Bhart-Anjan Bhullar, the Bhullar
Lab, the Briggs Lab, Travis Brady, Sloane Smith (Yale University), Eva Herbst, Krijn Michel,
Andrew Cuff, Peter Bishop, and Louise Kermode (Royal Veterinary College) for advice and

assistance. We also acknowledge the roof slate miners who discovered many of the Hunsrück Slate
fossil specimens.

**Ethics.** We used no live animals in this study. The fossil specimens are housed by the Goldfuß
Museum of the Steinmann Institute of the University of Bonn and were not collected as part of
this investigation.

**Funding.** This project was funded by the National Science Foundation (NSF Award 1701830),
the Yale Institute for Biospheric Studies, the Paleontological Society, the Silliman College George
Shultz Fellowship and the Pierson College Richter Fellowship. 24

**Data accessibility.** The micro-CT scans are available from FigShare:
<https://figshare.com/s/ad6f1a971cece0a8827b>

**Competing interests.** We have no competing interests.

**Authors' contributions.** EGC, JRH, and DEGB were involved in conceptualization and funding
acquisition. EGC processed the micro-CT data. EGC and JRH constructed the digital models.
EGC, JRH and DEGB interpreted the results and prepared the manuscript.

**References**

[revised manuscript text omitted]

40. Clark EG. 2019 Ophiuroid locomotion from fundamental structures to integrated systems.
*Zoosymposia* **15**, 13-22.
41. Spencer WK. 1930 British Palaeozoic Asterozoa. Part 8. *Palaeontographical Society*
*Monographs* 1928, 389-436, plates 25-28.
42. Jell PA, Theron JN. 1999 Early Devonian echinoderms from South Africa. *Memoirs of the*
*Queensland Museum* **43**, 115-199.
43. Hutchinson JR, Anderson FC, Blemker SS, Delp SL. 2005 Analysis of hindlimb muscle
moment arms in *Tyrannosaurus rex* using a three-dimensional musculoskeletal computer
model: implications for stance, gait, and speed. *Paleobiology* **31**, 676-701.
44. Ziegler A. 2019 Combined visualization of echinoderm hard and soft parts using
contrast-enhanced micro-computed tomography. *Zoosymposia* **15**, 172-191.
45. Paschal T, Bell MA, Sperry J, Sieniewicz S, Wood RJ, Weaver JC. 2019 Design,
fabrication, and characterization of an untethered amphibious sea urchin-inspired robot.
*IEEE Trans. Robot. Autom.* **4**, 3348-3354.
46. Kano T, Kanauchi D, Aonuma H, Clark EG, Ishiguro A. 2019 Decentralized control
mechanism for determination of moving direction in brittle stars with penta-radially
symmetric body. *Front. Neurorobotics* **13**, 1-7.
47. Kano T, Kanauchi D, Ono T, Aonuma H, Ishiguro A. 2019. Flexible coordination of
flexible limbs: Decentralized control scheme for inter- and intra-limb coordination in brittle
stars' locomotion. *Front. Neurorobotics* **13**, 1-11.

Figure 1. (a-c) *Euzonosoma tischbeinianum* (EGR 27), (d-f) *Ophiurina lymani* (HS 705). (a,d) Photograph, note that EGR 27 has undergone minimal preparation. (b,e) X-ray image. (c,f) Digital rendering of the 3D micro-CT image in Autodesk Maya, (c) shows only one arm of EGR 27.

244x176mm (144 x 144 DPI)

Figure 2. Arms of (a) *Encrinaster roemeri* (Hubo 116), (b) *Euzonosoma tischbeinianum* (EGR 27), (c) *Loriolaster mirabilis* (ESCH 596), (d) *Cheiropteraster giganteus* (Hubo 119), (e) *Furcaster palaeozoicus* (OKL 96), and (f) *Ophiurina lymani* (HS 705). Views of ventral (left) and dorsal (right) surface of arm and details of segments on ventral (above) and dorsal (below) sides (dorsal and ventral details of *Furcaster palaeozoicus* and dorsal details of *Loriolaster mirabilis* are from a different arm of the same specimen). Arrows indicate depressions corresponding to podial basins in *Ophiurina lymani*.

244x157mm (144 x 144 DPI)

Appendix B

RSOS-200329 Response to Reviewer Comments

We are grateful to both reviewers and the editor for their prompt review of our paper, and for their constructive comments which have substantially improved the manuscript. We have made the vast majority of suggested edits, with only a few exceptions. Comments from the reviewers are numbered below, with our response inserted beneath each one. Text that has been added to the main document is in red. Line numbers refer to the final submitted .docx version of the manuscript (please note that line numbers may have shifted slightly in the PDF compilation for review).

Reviewer 1

Reviewer #1 offered many insightful comments, which include suggesting that we unify the terminology used and expand on the insights offered by our new data.

1. There is a considerably diversity of morphological terminologies used for Paleozoic ophiuroids. I saw that you cited references in some parts of your descriptions. For a paper that focuses on function and morphology, however, it would be more advisable to pick an existing terminology, preferably one that makes sense in the context of a functional interpretation of the skeleton and provide the relevant references in the materials and methods section. Alternatively, if you feel the existing terms preclude a meaningful interpretation of function, you could come up with a new terminology. Either way, there is a general need of clarification in this respect.

Changes made: We now adhere to the terminology used by Glass and Blake (2004) and Gladwell (2018).

2. Regarding the conclusions of your paper, I agree with you that there are fundamental differences between the taxa you analysed in terms of arm structure. I also agree that these differences most probably reflect contrasting locomotion modes. However, I cannot see in what way the 3D digital renderings provide new insights. I do not question the interest of such approaches but I feel their surplus evidence is not conveyed in a convincing way here. For example, it was clear from previous works that ophiuroids with alternating ambulacrals most probably employed podial walking. You even cite references that drew that conclusion from their own morphological observations, so I cannot see in what way your data provide new insights here.

Changes made: Our intention is not to underplay the importance of other methods for interpreting the morphology and function of Palaeozoic ophiuroid arms. The advantage of scanning is the potential it offers to image all the details of a complete specimen in 360° and to explore function in an analogous fashion to that applied to living ophiuroids (e.g., Clark et al. 2018, *Journal of Anatomy*). One of our aims is to encourage and enable future researchers to investigate the Hunsrück Slate asterozoans anew, taking advantage not only of a new method for visualizing specimens, but also capitalizing on new material accumulated in recent decades.

With a few notable exceptions (e.g., papers by Glass and Blake) we are still largely reliant on the investigations of Spencer (completed by the 1950s), and Lehmann (1957) for information on these fossils. We have shifted the text to emphasize the advantages of this methodology and the novel insights revealed through its application here. We have also added text to specify the new data and insights that can be obtained from employing this technique (e.g., line 67-78, 333-343).

3. You quite correctly stress that many critical details are undetectable when observing articulated skeletons in dorsal and ventral views only. Reports of dissociated skeletal parts (e.g. Boczarowski 2001), however, provide exactly that type of detail, e.g. presence or absence of ball-and-socket articulations. So again, I cannot see how your data provide new insights here.

Changes made: We now cite Boczarowski (2001) (line 307). Scans are important, however, because complete specimens are critical to the interpretation of total morphology: visualization of the articulations between successive elements are necessary to understand their integrated biomechanical function. Isolated ossicles are not available for all taxa, and the identification of isolated ossicles in the absence of complete specimens can be difficult (e.g., 261-277, 326-348).

4. I see a fundamental problem with conclusion in lines 340 - 342: how can you say that "the expression of musculoskeletally driven locomotion occurred after the evolution of opposing ambulacrals, but preceded the fusion of the ambulacrals to form vertebrae" when you analyse only taxa with ambulacrals either alternating or fused into vertebrae? Furcaster has the ambulacral halves not only opposing but firmly fused into vertebrae, much like the living ophiuroids. It differs fundamentally in articulation type and in the position of the water vessel canal, and I would have loved to see if these differences matter in locomotion.

Changes made: This is a fair comment. Our inference is reasonable and supported to some extent by the phylogenetic analyses of Shackleton (2005) and Glass (2005, 2006), but these phylogenies do not include all the taxa we imaged. We have rephrased the text accordingly (line 333-348, 376-383).

5. As for Ophiurina lymani, I do agree that there are some similarities with Amphicutis stygobita but there are many other living species with similar arm morphologies that do not rely on podial walking (e.g. most ophiomusaid and ophiosphalmid taxa). The position of the podial basins is not unusual at all since almost all ophiuroids have their tube feet at the distal ends of the arm segments.

Changes made: We have revised the text to reflect the equivocal nature of the evidence (lines 124-136, 359-369).

Reviewer 2

Reviewer #2 offered many helpful comments, including suggesting that we emphasize the benefit of our chosen methodology and clarify the basis of certain statements in the discussion.

1. Line 59: I am afraid this is not the case, this might be true for Hunsrück Slate but not for the vast majority of specimens in museum collections. To say that their methods reveals both dorsal and ventral surfaces of each specimen for the first time together with other previously unobserved characters such as the proximal and distal surfaces of the arm segments is not true.

Changes made: We now specify that this statement applies to the fossil brittle stars of the Hunsrück Slate and we have clarified the potential advantages of our approach (line 165-173, 397-401).

2. Line 129: These are in effect morphological descriptions that use classification *sensu* Glass & Blake without mentioning any of the background of the terminology used. In fact, at best these are half way house morphological descriptions and CT scans. It read more like a brief guide to taxa. It does not include background from existing extensive literature on these taxa by Spencer, Blake or Glass. The descriptions have lots of basic errors that demonstrate a general ignorance of the systematics of the group.

Changes made: It is not clear how we should interpret this and the following comments with specific reference to Line 129. However, we have carefully reviewed the text, adding citations as appropriate and making changes to ensure that our use of terminology is both consistent and correct.

3. Line 129: In addition, there is no discussion on the influence of taphonomy on the morphology of these ophiuroids.

Changes made: We have included statements indicating where the quality of information is compromised by the quality of preservation (e.g., line 281-282, 295-297, 314-315).

4. Line 129: Recent papers on the fauna of South Africa have not been included.

Changes made: We added references to Reid et al. 2015 and 2019 on the Bokkeveld Group, and to Fraga and Vega (2020) which treats material from south Brazil, but with extensive mentions of South African occurrences, mainly with reference to the views of Jell and Theron (1999) (e.g., line 72-73, 140-142, 269).

5. Line 304: The discussion simply repeats observations that have been known for a long time and can be observed in Spencer, Blake or Glass. For we know that “ossicles of many Palaeozoic ‘archaic’ ophiuroids, are paired and offset along the axis of the arm”. It is also well known that the arms are largely incorporated into the body disc.

Changes made: This discussion is included to provide necessary context. We certainly did not intend to give the impression that these were original observations (as Reviewer 1 states, they are well documented in the literature). We have now added appropriate citations throughout this section.

6. Line 304: The observation that “there are no extant animals that use two offset columns of repeating segments for musculoskeletally driven locomotion is very interesting but I don’t understand from the MS how this influences movement.

Changes made: Every animal that engages in musculoskeletally-driven locomotion, from insects to vertebrates, applies rotational motions about a joint interface. This investigation is the first to observe the full morphology of the articulations between the skeletal elements within the arms. We found that Paleozoic brittle stars with two offset columns of repeating segments lack distinct joint interfaces; thus, we can infer that they did not exhibit musculoskeletally-driven motion analogous to that used by extant animals. We elaborate on this topic in the text (lines 326-348).

7. Line 313: It is a very bold claim to say that archaic ophiuroids (Encrinasteridae, Cheiropterasteridae) did not use musculoskeletally driven locomotion similar to that in modern ophiuroids. So they must have relied on their tube feet for locomotion (‘podial walking’). I don’t see the logic behind this assumption. Why could they have moved their arms? In addition, they have not taken into account that many parts of the ophiuroid skeleton are decalcified. The authors have not taken into account the soft tissue structures that could have existed.

Changes made: Please see our response to comment 6 above. We never claimed, and do not believe, that Paleozoic brittle stars would not have been able to move their arms. In fact, we argue for a specific strategy for locomotion employed by these forms in lines 376-383. We outline the logic that we use to support this strategy in lines 326-348. We certainly have taken into account soft tissue structures that most likely existed in the arms of Paleozoic ophiuroids. As in modern ophiuroids, it is reasonable to infer that Paleozoic ophiuroids with alternating rows of ambulacrals would have had muscle and collagenous tissue between successive elements regardless of the form of locomotion employed. In fact, we even suggest locations of potential muscle attachment sites in these taxa (lines 223-224). We are not familiar with any evidence that “many parts of the ophiuroid skeleton are decalcified,” and are not aware of any non-calcareous skeletal tissue in extant ophiuroids.

8. Line 313: It is interesting that the more derived *Furcaster palaeozoicus* among the Palaeozoic ‘archaic’ ophiuroids is the most similar to that in typical extant taxa with bilaterally symmetrical segments, ball-and-socket joints, resembling the structure and location of these joint interfaces and muscle attachment sites. However, this is rather a large jump in logic without a functional test of how both these arms functioned. This makes the discussion highly speculative.

Changes made: We have eliminated reference to an 'archaic' group in the text due to the confusion around this term as suggested by Reviewer 3. We describe the features noted by the reviewer here while eliminating text that overstepped the inferences that can be made from the morphology at this time (line 333-358).

9. Line 326: Why does the morphology of the arm segments of *Ophiurina lymani* provide compelling evidence for podial walking? It is a bit of a jump in logic to say that they are remarkably similar in shape to those of *Amphicutus stygobita*, the extant cave-dwelling brittle star and thus must be a tube feet walker. So many other ophiuroids have this morphology and are not tube foot walkers!

Changes made: This was also raised by Reviewer 1, comment 5. We have revised the text to reflect the equivocal nature of the evidence (lines 124-136, 359-369).

10. Line 332: The most important result of this study is the evidence of disparity of locomotory function in the pyritized Hunsrück Slate brittle stars. However, as warning, details of the articular surface of the vertebrae of living brittle stars do not necessarily correspond to differences in locomotory capability. I think it is true that examining 3D morphology reveals striking contrasts in the morphology of the arms of Encrinasteridae + Cheiropterasteridae, *Furcaster palaeozoicus* and *Ophiurina lymani*, which imply significant differences in function. It is clear there is an evolutionary sequence of development in the arms moving from stepped ambulacrals and inline ambulacrals which is very interesting. But why did you choose the Hunsrück Slate brittle stars when this transformation was very likely to have taken place by the late Silurian? This is in mind it is simply not possible to state that musculoskeletally driven locomotion occurred after the evolution of opposing ambulacrals, but preceded the fusion of the ambulacrals to form vertebrae, or that rows and columns of opposing ambulacrals must have been sufficiently articulated to exhibit this form of motion when you do not have enough data (only 5 taxa) to test this transformation, which might have already taken place by the Devonian.

Changes made: We selected Hunsrück Slate material for several reasons. 1. The density of the material provides an appropriate contrast to conduct an exploratory analysis of the advantages of using micro-CT scanning to examine fossil ophiuroid anatomy. 2. The diversity and disparity of ophiuroid taxa within this locality provide an opportunity for a rich comparative analysis. 3. Limiting the investigation to the Hunsrück Slate allows the morphology of each specimen to be directly comparable in terms of their preservation (lines 162-175).

Line 357: Finally it's not correct to say "Prior to this study, the 3D morphology of Palaeozoic brittle star arm ossicles was mainly inferred by integrating the morphology of their dorsal and ventral surfaces exposed by preparation of different specimens." Such information is already available from many other well preserved specimens using traditional methods.

Changes made: We have clarified that we are referring specifically to the ophiuroids of the Hunsrück slate here (line 395-397). We also clarify the specific advantages of 3D imaging for revealing previously inaccessible morphological information, such as the articulations between successive segments and the full morphology of a single arm in the round for the first time (line 39-40, 73-78, 165-173, 395-401), which provided critical new functional insights (e.g., 261-277, 326-348, 376-383, 401-404).

Reviewer 3

Reviewer #3 offered many detailed suggestions, including providing further explanation regarding the methodology, modifying the terminology used, and clarifying some of the taxonomic information provided.

Line 35: I strongly advise AGAINST using this term. I believe it was first used to refer to Paleozoic ophiuroids without vertebrae by Hunter and MacNamara 2017. They reference Spencer (1951) but he never used the term, and the groups he (and they) included in this are not all ophiuroids. Indeed, Hunter and MacNamara (2017) seem to use the term in lieu of a formal clade name for the groups they include in it. Unfortunately, the term has been adopted by others, despite the fact that it is poorly and inconsistently defined. It also "reeks" of that former teleological burden that was "primitive" versus "advance". This might not be as obvious in the English language but other languages consider "archaic" functionally synonymous with terms like "lesser", "not as good", "outdated" etc. Btw, as used by Hunter and MacNamara (2017) the term "archaic" as you use it, is not the same. "Ophiurina" and "Furcaster" are part of the "modern" group (with vertebra), as they use the term. I think you should cut it and use "Paleozoic" instead (the debate whether "Furcaster" existed into the post-Paleozoic might just become a citation such as (but see...Thuy...) here).

Changes made: We have abandoned the use of 'archaic' and revised the text accordingly.

Line 38: By "structure", do you mean "shape of the ossicles"? or "shape of the overall arms"?

Changes made: We changed this sentence to "unique morphology and arrangement of multifaceted skeletal elements..." (line 31-32).

Line 38: You want to be careful in your wording here. Physically prepared specimens are also in 3D, that is they have form and shape outside of 2 dimensions. To dismiss prepared specimens of Hunsrueck Slate as being two-dimensional is an overstatement at best and wrong at worst. Yes, there are flattened specimens in the Hunsrueck but there are plenty who are incredibly three-dimensional (I think, see images in Briggs et al 1996 for example)

Changes made: We have changed this sentence to "Our digital approach allows the structure of the arms of specimens of these taxa to be visualized *in situ* in the round for the first time." (lines 39-40).

Line 40: maybe, but of course without a phylogeny we do not know whether "type of locomotion" is a shared trait or has evolved multiple times in separate lineages.

Changes not made (no changes necessary): We agree, and is why we have not speculated as to the monophyly or paraphyly of these traits in the text.

Line 46: Citation here?

Changes made: We have added a citation specific to brittle stars here (line 50).

Line 49: "Closest relatives" should be "closest living relatives."

Changes made: We have made this change as suggested. (line 51).

Line 51: This isn't controversial but I do wonder, are you making the argument that ophiuroids evolved from asteroids and therefore from a tube-foot walking ancestor? Trouble is, that somasteroids are a much more likely ancestor of ophiuroids - they too have large podial basins and PRESUMABLY were tube-foot walkers, but you need to make sure that you are using the asteroids as a functional analog, NOT as an ancestral group to ophiuroids.

Changes made: We don't argue that ophiuroids evolve from an asteroid ancestor; rather, that the lineage including ophiuroids likely utilized tube-foot walking based on the expression of this form of motion in the most closely related extant classes. We have changed this to "including the sea stars and sea urchins" (line 52).

Line 60: Careful here. You don't want to set up a "straw man". Yes, in some instances we have SINGLE specimens exposing only ONE side. But many fossil ophiuroid taxa are known from multiple specimens, each showing different sides and aspects of the animal. Furthermore, preparation can, and has often, been done from both sides on the same animal (even in the Hunsrueck, for example in older specimens at the British Museum)... I understand that you are trying to make a strong argument for the use of microscanning here but I DO think you are overstating the case here a little bit when it comes to the "limitations" of other methods.

Changes made: We have changed this text in light of the reviewer's comment (line 64-78).

Line 62: I am not sure that these approaches consistently "fail to reveal critical details". They do when the specimen is fully articulated and spacing between ossicles is narrow. There are countless examples however of specimens that are partially disarticulated (including in the Hunsrueck) that reveal these features.

Changes made: The main advantage of using micro-CT scanning in terms of biomechanical observations is to view the articulations between successive elements *in situ* in 3D. This is not possible to do using traditional methods. We have added a sentence to this effect in line 75-78.

Line 64: Oddly honest. Are you saying that your paper included "errors of interpretation"? Which ones? I think your 2017 paper makes the case FOR microscanning, no? Now you might be referring to your disagreement with Glass and Blake (2006) whose "interambulacral muscle gaps" you reinterpret as actual ossicles A and structure B. However, to other ophiuroid workers, ossicle A is clearly the dorsal surface of the ambulacral, whereas structure B represents pyrite or sediment filled gaps. These interpretations are far from unequivocal, indeed they create uncertain based on the three-D scans rather than more agreement on the morphology of these animals, no? So as, amazing and useful as scanning is, it brings with its own ambiguities. BOTH preparation AND scanning can leave room for error.

Changes made: We have edited the text to clarify our meaning here (line 75-78).

Line 70: ...traditionally by the influential treatise" "Traditional" by itself is a little vague. There were many different interpretations of ophiuroid groups before the treatise.

Changes made: We changed this sentence to "have been recognized..." (line 81). Here, the intention is not to provide an exhaustive overview of the history of fossil ophiuroid taxonomy; rather, our aim is to summarize the current taxonomic state of the group and connect currently accepted taxa with nomenclature used in the recent past.

Line 80: Hm. Probably want to add here the work of Hotchkiss et al. 2007, and Hotchkiss and Glass (2012), which showed that the fossil record of Paleozoic ophiuroids is, well, non-existent. So well before the molecular work of the authors you cite here, paleontologist concluded that there were no Paleozoic euryalids. That deserves credit.

Changes made: We have added a citation to Hotchkiss and Glass (2012). Hotchkiss et al. (2007) provide a less direct statement of this conclusion (line 91).

Line 83: see my comment above, also is this the case for ALL ophiuroid workers today? I am not sure. Are there biologist who still think that ophiuroids evolved from echinoids?

Changes made: The most recent phylogenetic analyses support Asterozoa (references 2-3). This paragraph provides references for interpretations of the evidence regarding the evolutionary history of Paleozoic ophiuroids. We have changed the first sentence to "Ophiuroids have been hypothesized to have evolved from a somasteroid ancestor." (line 94).

Line 86: See comment above about which taxa you included in archaic which actually have vertebrae

Changes made: We have eliminated reference to the notion of an 'archaic' group in the text.

Line 88: it is important to recognize that this outcrop was chosen due to the availability of well-preserved material, NOT because it was deemed to be basal to modern ophiuroids (indeed, Shackleton seems to have shown that this is not the case)

Changes made: This text has been removed from the manuscript. We have added text regarding our choice of locality (162-175).

Line 93: Depending on the guidelines of the journal, you might include some of the work by Glass here. Although his phylogeny remains unpublished, his work has been cited as such for completeness sake by other authors in the literature. I think there is an abstract out there that summarizes the phylogeny of Glass. Until it is published, of course, it is up to you whether to include it or not.

Changes made: We have included reference to this investigation (line 190-191).

Line 104: There are vertebrae-bearing ophiuroids, such as Furcaster, prior to the Devonian. Also look at discussion on Hallasteridae by Spencer. Again, your use of "modern" versus "archaic" is inconsistent (I know you are using it because others have unfortunately done so, but it is not consistent here). I suspect by modern arm morphology you mean vertebrae PLUS dorsal and ventral arm ossicles, radial shields, etc. But in sentence 101 and 102 you define modern as "fused vertebrae" - well those are present well beforehand.

Changes made: We have dealt with this by abandoning the reference to an 'archaic' group throughout the text.

Line 107: You need to make a careful search through publications by Hotchkiss in this regard. He has made some very important and original contributions to the discussion of the possible connections between stem and crown group ophiuroids - start with Hotchkiss and Haude (2004) work on Aganaster and Stephanoura, for example. Unfortunately, much work on Paleozoic "ophiurids" has either been ignored or downplayed in significance by recent neontological papers on the subject. You do NOT want to make the same mistake.

Changes made: This point is made by Thuy et al. (2015) and we now cite that paper and Hotchkiss and Haude (2004) (line 108-111, 117-119).

Line 116: So does "Ophiogeron supinus", see Byrne and Hendler (1988) and discussion thereof in Glass and Blake (2004).

Changes made: We have added a comment on this taxon and cited these references (line 133-136, 364-369).

Line 118: See my comment above - not sure one follows the other the way you set it up here.

Changes made: Please see our response to this reviewer's comment regarding Line 51.

Line 121: As used here that would exclude Ophiurina lymani...

Changes made: We have rephrased the text.

Line 124: might also add Hunter et al 2016, and Fraga and Vega 2020 here, for completeness sake. These authors have also discussed this and provided important diagrams.

Changes made: We have added both references (line 140-142).

Line 138: by "morphology" do you mean, the overall shape of the arm, or the articulation surfaces and muscle attachment sites of the individual ossicles. I think the latter, no?

Changes made: Here, we are referring to the emergent structure generated by the shape of individual ossicles and their positions relative to one another. We have clarified this in the text (lines 155-157).

Line 139: Some people limit this term when talking about soft-tissue preservation. Might "well-preserved, fully articulated" specimens be more appropriate?

Changes made: We have made the changes as suggested (line 157-159).

Line 145: Careful, a common misconception in the literature is that MOST or ALL specimens in the Hunsrueck Slate are fully articulated and completely preserved. This is NOT the case. The opposite is true. However, it does contain a noteworthy number of completely preserved specimens.

Changes made: We have modified the statement to reflect the reviewer's observation (line 163-165).

Line 147: Glass and Blake (2004) provide a discussion of this interpretation and raise some doubts about the nature of the tracemaker.

Changes made: We have added a citation to Glass and Blake (2004) but we omit a discussion of the trace maker here as it would be a digression from the main topic of the paper (line 165).

Line 149: and entire ossicles! Delicate spines for example.

Changes made: We have updated the text to this effect (line 167).

Line 150: need to include Bartels et al. 1998 here - they provide the most comprehensive summary of these methods, Glass and Blake (2004) only stand on the shoulder of these giants.

Changes made: We have updated the text with this reference (line 168-169)

Line 150: Can you provide a timeframe for both airbrasive and 3d-scanning (including analysis in the latter, travel to available equipment) etc. Since you are selling microCT scanning as something superior it would help to compare all aspects of these two techniques.

Changes made: We have added a reference to Bartels et al. 1998 (line 168-169) and Stöhr et al. 2019 (line 173) which detail the airbrasive method and micro-CT scanning respectively for analyzing ophiuroid morphology (line 168-173).

Line 151: What is the density difference?

Changes made: Micro-CT scan technology relies on a contrast in density between the fossil and matrix but does not necessitate specific calibrated values in this regard. We have rephrased to emphasize the contrast without qualifying it (line 169-171).

Line 156: Any particular reason for this? Are these particularly well preserved? Complete? Of historical significance? Readily available? Why not the British Museum or Berlin collections?

Changes made: This Steinmann Institute of the University of Bonn provided access to one of the largest collections of Hunsrück Slate ophiuroids and ample machine time to x-ray image and micro-CT specimens. The facility for X-ray screening on site was a particular advantage in avoiding expensive CT scanning of unsuitable specimens. This facility was also selected due to the Hunsrück Slate expertise of the scientists associated with the Steinmann Institute. We have added text to clarify these advantages (lines 176-178).

157: "quality of preparation" - I thought you said this was done in lieu of preparation. So where some of the specimens already prepared, and if so, which methods. Did the CT method bring out more details?

Changes made: This text refers to eliminating specimens from consideration in which previous poor preparation removed critical details. We clarify this in lines 178-180.

Line 157: what is the size limitation of the scanner - this might be useful information for those who plan to do similar work

Changes made: Size limits vary depending on the resolution desired, the shape and thickness of the slab and placement of the fossil within the specimen. We direct the readers to several papers on using 3D imaging to observe morphology in ophiuroids in the text (lines 203-204).

Line 159: percentage? Might be useful information to show how important it is to x-ray fossils first.

Changes not made (no changes necessary): Suitability for micro-CT scanning is a gradient rather than a threshold and the selection is based on criteria that vary between specimens (completeness, preparation etc.) and from species to species. Thus a specific percentage would not be meaningful.

Line 161: So some specimens WERE already prepared using traditional methods?

Changes made: Yes, and this is now clarified in the text. Most of the ophiuroid specimens within museum collections have been prepared to a certain degree, not least to enhance and identify them as they are normally concealed to some extent by matrix. Nonetheless our results indicate that, in most cases, micro-CT scanning with minimal preparation is the best way to reveal critical details. Information regarding preparation is provided throughout the text (e.g., 193-195, 280-282, 295-297, figure 1).

Line 163: segments?

Changes made: Yes, as was the case for HS 705 (line 314-315). We changed the text to read “ossicles (or segments)” (line 185).

Line 165: so out of the 33, you chose 6 for CT scanning.

Changes made: Yes. We have clarified by modifying the text in lines 187-192.

Line 167: What do these numbers mean? Are they specimen numbers? Different collections?

Changes made: These are specimen numbers; we have clarified this in the Data Accessibility statement (line 449-452).

Line 168: Did Shackleton use Protasteridae for Furcaster? Why not call them Furcasteridae?

Changes made: Shackleton did assign *Furcaster* to Protasteridae. However, Glass (2005, 2006) and Hunter et al. (2016) assigned it to Furcasteridae. We have changed the family name to Furcasteridae in line 190-191 and 300, and noted that *Furcaster* was previously assigned to Protasteridae in line 190-191.

Line 170: or completely!

Changes made: We have added text to clarify this (lines 193-195).

Line 176: but these did not include Hunsrück specimens, yes? So this IS the first time this has been done with Hunsrueck?

Changes made: We now state that this is the first time this technique has been applied to analyze the ophiuroids of the Hunsrück Slate, to our knowledge (line 204-205).

Line 180: Again, it might be useful to say something about the time involved in "extracting" and "cleaning" since you claim it is faster than conventional preparation techniques.

Changes made: We have added a sentence with additional information about the techniques used and directed the reader to a methods paper where the methods used are described (lines 203-204).

Line 189: After reading through this I was struck by the fact that the observations in red have already been made for these taxa by other authors, even those who first described these taxa like Roemer, Stuertz, and Lehmann. Since you want to sell CT scanning as a novel method that yields novel results (arguably previously inaccessible through conventional methods), I would recommend you clearly separate these novel observations from those that have already been made by others, and/or can be readily observed through study of specimens in x-ray, mechanical, and airbrasive methods (say for example the proximal and distal surfaces of the individual ossicles).

Changes made: Please see our response below to this Reviewer's comment regarding line 315. We have also added text regarding the specific features of Hunsrück ophiuroids that are currently only accessible for observation through 3D imaging (line 75-78).

Line 198: I am not sure what this refers to. The spaces that used to be filled with muscles?

Changes not made (no changes necessary): Likely yes; however, we prefer not to speculate regarding function in this section.

Line 203: Can you be more specific here? Wouldn't ambulacral shape and podial basin size simply change because of where they are in the arms, in other words not exactly a surprising discovery.

Changes made: This statement refers to the relative size of these features compared to other dimensions of the ambulacrals along the length of the arm, not the absolute size. We found that various characteristics of each species scale differently along the length of the arm; we clarify that we refer to the relative size of these features (line 230-231).

Line 207: Interesting. What are these? Are they ossicles like sublaterals?

Changes made: We discuss these elements further in lines 261-268 within the description of *Euzonosoma tischbeinianum*.

Line 208: can you mark these on the images?

Changes made: We have annotated these features in Figure 2 (line 419-422).

254: or degree of pyritization?

Changes made: We have added text explaining that It is unlikely that this is due to degree of pyritization because it is only evident in the area of the specimen that has been prepared (line 280-282).

256: archway - need to be careful with this descriptors. Please see Blake (2019) on whether ambulacrals are arched, vaulted, etc. across different stelleroids. See his nomenclature.

Changes made: We are not describing the ambulacrals *per se* but the passage they create for the radial canal.

262: Hotchkiss et al. 1999 already described many, if not most, of these features of the ambulacrals of cheiropterasterids.

Changes made: We now cite this paper in the description of *Cheiropteraster* (line 291, 295).

266: So the radial canal here passes BETWEEN ossicles rather than through the ossicle into the podial basin?!

Changes made (no changes necessary): In both Paleozoic and extant ophiuroids, the radial canal passes between ossicles; the lateral canals pass through the ossicles into the podial basins.

268: Yet, this is what you said CT scanning could help with...

Changes made: This is true but it is not due to a limitation of the method but because of inadequate pyritization of the elements. We have clarified this (line 295-297).

283: do you mean the "wings" as in Haude and Thomas 1983?

Changes not made (no changes necessary): No, the wings (Flügel) are on at the distal end of the segments in the species of *Furcaster* illustrated by Haude and Thomas (1983).

286: Again, isn't this how you sold CT scanning at the beginning of the paper? I don't doubt that CT scanning can do this but it is more nuanced isn't it?

Changes made: This isn't due to a limitation of the method; rather, to the pyritization of the specimen. We clarify this (line 314-315).

306: above include Furcaster and Ophiurina in your list of "archaic" ophiuroids.

Changes made: In response to the advice of this reviewer, and in spite of its recent use by others, we have abandoned the term 'archaic'. Here we have clarified that we are referring to

taxa in which the arm ossicles are paired and offset: *Furcaster* and *Ophiurina* do not fall into this category (line 334-336).

307: I think this statement cannot be justified based on the evidence provided herein. It is not clear from the images whether these joint faces are merely close together and therefore obscured or whether they are lacking altogether.

Changes made: This investigation provides unequivocal evidence that joint interfaces are absent between successive elements in the columns of ambulacrals in *Encrinaster roemeri* and *Euzonosoma tischbeinianum*. This is clearly evident in the micro-CT scans, the 3D models, and the figures included in the text. We emphasize this in lines 336-341.

307: ambulacrals?

Changes made: We have changed “ossicles” to “ambulacrals” here (line 339).

310: Which would make arm coiling pretty much impossible.

Changes not made (no changes necessary): Although the incorporation of arms into the body disc presumably limits the degree of flexibility, determining the range of motion permitted by the skeletal morphology is outside the scope of this investigation.

310: also see discussions in Spencer (1922, 1950, and 1951)

Changes not made (no changes necessary): Spencer (1950) is concerned primarily with distribution through time. Spencer (1922, 1951) offers similar comments on mode of life to Spencer and Wright (1966) but not with respect to the genera considered here. The citation to Spencer and Wright is therefore appropriate.

312: This is an odd statement. Among echinoderms? Are there animals OTHER than echinoderms with "two offset columns"? This analogy and reasoning from it isn't clear.

Changes made: No- there are no animals, to our knowledge, in the animal kingdom with two offset columns of skeletal elements used for musculoskeletal-driven locomotion. We have rewritten the text in this section to clarify and emphasize this point (lines 326-348).

315: These are not novel conclusions or interpretations for these taxa and it is not clear how new findings by the CT scanning above was used to derive these conclusions. Please take a look at Harper and Morris (1978) and their discussion of functional morphology of encrinasterids in general and the cheiropterasterid *Armathyra*, as well as discussions about arm mobility and tube foot walking in Byrne and Hendlar (1983).

Changes made: We have substantially modified the text throughout the manuscript by adding additional citations to previous literature, emphasizing the specific visualization capabilities

only possible through the methodology utilized, highlighting the novel aspects of our new data, and modifying the interpretation of our results through the suggestions of the reviewers. We fundamentally disagree with the notion that our findings are not novel, as ours is the first application of the only known technique in which ophiuroid arm morphology can be observed in 360° in complete fossils and the articulations of individual skeletal elements with one another can be observed; it is not possible to collect these data by traditional means. A strength of our paper is that we have a stronger understanding of the mechanics of motion in living ophiuroids and the degree to which functional inferences can be realistically gleaned from morphological observations (based on Clark et al. 2018). This approach will allow future testing of the kind of functional interpretations offered by Harper and Morris (1978), for example, that were based on the data and methods available to them at the time. In this text, we identify practical avenues for functional analyses using novel visualization and modeling techniques moving forward. We have added citations to the literature that the reviewer suggests (line 217, 345).

317: again, so what is "archaic" mean then?

Changes made: We have abandoned this terminology.

324: some?

Changes made: We have made the change suggested by the reviewer (line 359).

331: These comparisons are interesting and possible. However, Ophiurina, as an ophiurid, shares all sorts of characteristics with post-Paleozoic ophiuroids, not the least of which are flexible arm segments made up of vertebrae. Would the two tube feet in each segment really be sufficient to "lift this animal off the substrate to walk on them? Or, did they use them while also using their flexible arm to crawl around? There is nothing here or elsewhere in the literature to suggest that Ophiurina was weak muscled, lacked ball and socket joints, or could not use its thin flexible arms for "normal" ophiuroid movement. Of all the ophiuroids discussed here, one might argue Ophiurina is the most "modern" and therefore the most likely to exhibit a similar mode of locomotion, right?

Changes made: We have revised the text to reflect the equivocal nature of the evidence (lines 359-369).

342: I am not sure how this statement can be justified based on the evidence presented herein. Plus, it has been widely hypothesized and discussed by many other fossil ophiuroid workers, that the modern musculoskeletally driven locomotion was made possible by the formation of first opposite and then potentially fused amb. No ophiuroid with simple but unfused ambulacrals was included in this study....

Changes made: We have rephrased the text accordingly (line 349-358, 376-383).

361: This is simply not true. The amount of information provided by the Ct scanning on the nature, size, shape, and position of the articulation surface between ambulacrals remains equivocal based on the results presented here. Also, there ARE published accounts of what these articulations looked like in specimens from the groups herein, but not necessarily from the Hunsrueck, so CT might not be necessary to provide new insights on new interpretations of these.

Changes made: “Articulations” is not synonymous with “articular surfaces”; “articulations” refer to the position of the ossicles with respect to one another. There is no other non-destructive method available to visualize the complete morphology of the skeletal elements of fossil animals and their articulations with one another *in situ*. The method has yielded remarkable results even when applied to living ophiuroids (Clark et al. 2018, *Journal of Anatomy* 150-152, 330-331, 368-369). We clarify this in the text (lines 397-401).

401: As CT scanning is presented here as a potentially superior approach to conventional preparation, it would be prudent to include information on costs involved for each. Obvious limitation for airbrasive are cost of equipment, but cost of running scan, and software requirements for CT scanning might come with similarly high start up costs.

Changes made: Our assertion is that CT scanning has the potential to provide more information in a non-invasive fashion. A cost benefit analysis is beyond the scope of this paper and will vary based on equipment accessibility and institutional affiliation. We have added a reference to the “Material and methods” section (line 203-204) to a paper (Stöhr et al. 2019) that provides more information about this methodology specific to ophiuroids. As micro-CT scanning facilities become more widely available they may be easier to access than suitable airbrasive equipment.

Figure 2: These images are pretty small - it would help to have these be at least TWICE as big.

Changes made: We have divided Figure 2 into two separate figures, achieving the desired increase in magnification.

Appendix C

RSOS-201380 Response to Reviewer Comments

We are grateful for the opportunity to revise our manuscript, and to the reviewers and the editors for their constructive comments which have further improved the paper. We have made the vast majority of suggested edits with only a few exceptions. Comments from the reviewers are numbered below, with our response inserted beneath each one. Alterations to the main document are in blue. Line numbers refer to the final submitted .docx version of the manuscript (please note that line numbers may have shifted slightly in the PDF compilation for review).

Subject Editor Comments

1. On balance I think there is still more work to do and there is some concern about claims that yours is a "better" analysis when perhaps it is not bringing so much that is new. I don't weigh in on this but I hope that you can make its novelty clearer. Thanks and best wishes.

Changes made: Please see our response to Comment 1 by the Associate Editor.

Associate Editor Comments

The Associate Editor suggested that we clarify the novel insights provided by micro-CT imaging and describe how these techniques and findings contribute to the understanding of Palaeozoic ophiuroid locomotion. They also suggested a terminology change and that we address the influence of taphonomy.

1. There seems a general feeling amongst the reviewers that the championing of micro-ct scanning in the manuscript implies that previous methodologies are substandard, and ct-scanning has not revealed anything new that couldn't have been learned prior. While I agree with the reviewers that ct-scanning isn't a holy grail capable of solving all the problems of paleobiology, though after reading your responses to the first round of reviews I agree with you that it absolutely facilitates the kinds of hypothesis-driven biomechanical analyses that wouldn't be possible without 3D data. I would suggest that in the next round of edits, you explicitly stress that in order to carry out biomechanical analyses in the future, you need the kinds of data you have here. Perhaps instead of stating that 3d-visualizations shed new light by allowing for 360 visualization of the skeleton (Line 77-78), I might add a sentence or two stating that they allow for biomechanical analyses of animal function (with a few citations across animal groups). I think if this is a little more explicit, then the MS will live up to the expectations promised in the introduction.

Changes made: The reviewers, as well as both of the editors, expressed concerns regarding the novelty of the information provided by micro-CT imaging. To address these concerns, we now provide a more detailed explanation of the types of data that can be obtained using digital visualization technology that are not readily accessible with standard methods. We agree that critical morphological details of these taxa can and have been acquired using standard methods.

However, 3D geometrical data are required as the basis for certain biomechanical inferences that such methods cannot provide.

Motion in musculoskeletal-based organisms, and in synthetic devices using rigid structural components, relies on the application of rotational forces about a point at the intersection between two skeletal structures. Thus, to form reliable inferences about the production of motion in an organism with an internal musculoskeletal system, both the geometry of the surfaces of the skeletal structures and their 3D position with respect to one another must be known. Standard methods for investigating the morphology of both extant (i.e., dissections) and extinct organisms (i.e., fossil preparation) necessitate a trade-off between observation of individual elements and visualization of how they fit together. Although the surfaces of individual elements are typically obscured by soft tissue or fossil matrix, careful preparation can reveal key aspects of the morphology. However, it is impossible to view the surface area of internal skeletal elements while retaining their precise 3D geometric positioning relative to one another, particularly where they abut one another directly, because the point of articulation is obscured. Here the value of digital visualization technology is in providing the data required for accurate biomechanical inferences, as it is the only non-destructive technique available to date.

This explanation has been added to the manuscript (lines 114-119, 127-157, 374-382).

2. There also still exists a feeling amongst reviewers that nothing explicitly new regarding the locomotory capacity of Palaeozoic ophiuroids has been gleaned from these analyses. I don't personally believe that all new data need to change interpretations, and if these new data support previous interpretations, I think that doesn't preclude publication. I would thus suggest making it a bit more clear in the discussion and conclusions section when your results and interpretations are in agreement with those of previous authors (e.g. reference # 51).

Changes made: We have substantially revised the discussion to (1) review the hypotheses in the literature regarding the locomotion strategies used by Paleozoic ophiuroids, (2) address how our new data and other reports support the hypothesis that they did not use musculoskeletally-driven locomotion as in modern ophiuroids, and instead likely used podial walking, and (3) outline how micro-CT imaging can be used further as a critical tool to evaluate this hypothesis (line 325-382).

4. Additionally, the two new reviews obtained in this round (reviewers #4 and #5) have each requested some explicit changes, and some optional additions which I would suggest you incorporate into the MS before resubmission. In particular, I appreciate reviewer #5 mention that taphonomic artifacts, such as post-mortem disarticulation, aren't discussed much in the manuscript with regard to your interpretations.

Changes made: We have made almost all of the changes requested by Reviewers #4 and #5. We agree that taphonomic artifacts may have affected the *in situ* position of the skeletal elements. We minimized the influence of disarticulation due to taphonomy during specimen selection (lines 192-196), walked back the relevant conclusions in our revised discussion (lines 325-382) and outlined directions for future analysis of additional specimens which will minimize differences due to taphonomy (lines 275-278, 377-380).

5. I also agree with reviewer #5 about the use of “segments” in the manuscript. It’s a bit confusing, as I’m not sure if this related to some aspect of the morphology, or to “segmenting out” in making the 3D model. This clarification could help.

Although “segment” is the accepted terminology for the repeating modular series of arm ossicles found in living ophiuroids (e.g., LeClair 1996), we have changed “segment” to “ossicles” or “rows of ossicles” as needed when referring to Palaeozoic ophiuroids (e.g., line 185, 200, 316-321, 408). We use “extract” to refer to pulling out 3D data from the micro-CT scan (line 199, 201, 316)

Reviewer 1

Reviewer 1 suggested that we clarify the novel insights produced by micro-CT imaging and asked us to emphasize the importance of the new data provided. They also encouraged us to elaborate on how the new information provided advances the understanding of locomotion in Palaeozoic ophiuroids.

1. You rephrasing of lines 64-78 do little to solve the issue of Reviewer 3 - Line 60: “But many fossil ophiuroid taxa are known from multiple specimens, each showing different sides and aspects of the animal. Furthermore, preparation can, and has often, been done from both sides on the same animal (even in the Hunsrück Slate, for example in older specimens at the British Museum) I understand that you are trying to make a strong argument for the use of micro scanning here but I DO think you are overstating the case here a little bit when it comes to the “limitations” of other methods.”

Changes made: We have removed this text from the manuscript. In addition, see our response to Comment 1 by the Associate Editor and the text added to the introduction (line 114-119, 127-157).

2. The same is true for lines 75-78: which tried to address the comment of Reviewer 3 - Line 62: I am not sure that these approaches consistently “fail to reveal critical details”. They do when the specimen is fully articulated and spacing between ossicles is narrow. There are countless examples however of specimens that are partially disarticulated (including in the Hunsrück Slate) that reveal these features. I agree with reviewer 3 that it is not true that to view the articulations between successive elements in situ in 3D. This is not possible to do using traditional methods.

Changes made: We have removed the text referenced by the reviewer, clarified the advantages of micro-CT scanning, and emphasized the type of data that can only be obtained non-destructively using this technique (line 114-157).

3. In addition, I can’t see how changes to line (line 75-78) have addressed the concerns of Reviewer 3 - Line 64: that BOTH preparation AND scanning can leave room for error. The authors also use the same lines of text (line 75-78) to answer Reviewer 3 Line 189 concern

that observations you claim is novel have already been described by authors such as Roemer, Stuertz, and Lehmann. It appears you have made no effort to clearly separate these novel observations from those that have already been made by others, and/or can be readily observed through study of specimens in x-ray, mechanical, and airbrasive methods.

Changes made: We removed the text referenced by the reviewer. We have added text addressing the utility of micro-CT scanning for functional analysis (line 114-157). Our treatment of previous work is appropriate considering our text is focused on the application of micro-CT imaging, our manuscript addresses functional interpretation rather than taxonomic redescriptions, and the anatomical descriptions are limited to the specimens examined here. We cite relevant literature throughout the text, focusing on 20th century and later papers as these 19th century papers are cited in more recent taxonomic works.

4. I do not think the comments and changes you have provided address the serious concerns of Reviewer 3 Line. 315: “These are not novel conclusions or interpretations for these taxa and it is not clear how new findings by the CT scanning above was used to derive these conclusions. Please take a look at Harper and Morris (1978) as well as discussions about arm mobility and tube foot walking in Byrne and Hendler (1983)”

Changes made: We have added insights from and references to Harper and Morris (1978) and Byrne and Hendler (1983) (lines 216, 335). We have added text explaining how the new findings by the CT scanning were used to generate novel insights (line 114-157). We have substantially revised the Discussion, defining hypotheses regarding the locomotion strategies used by Paleozoic ophiuroids and the support provided by the new 3D imaging data here, as well as data from other reports (line 325-382, 393-395). We also provide information regarding how micro-CT imaging can be used as a tool to further address this hypothesis (line 145-157, 168-172, 194-195, 275-278, 374-382, 391-392, 399-401).

5. Reviewer 1 and 2 I cannot see in what way the 3D digital renderings provide new insights. I feel their surplus evidence is not conveyed in a convincing way here. For example, it was clear from previous works that ophiuroids with alternating ambulacrals most probably employed podial walking. You even cite references that drew that conclusion from their own morphological observations, so I cannot see in what way your data provide new insights here. Furthermore, it appears from the authors response down plays the work of Glass and Blake who significantly advanced the work of Spencer and Lehmann. I do not think future researchers need any encouragement to investigate the Hunsrück Slate asterozoans anew as we already have enough knowledge. A shift in methodology will only enhance our existing knowledge. But the existing knowledge and expertise needs to be reviewed and stated by the authors not overlooked.

Changes made: We have elaborated on the specific insights that can be obtained through micro-CT scanning, the only non-destructive method that reveals the 3D geometry, internal morphology, and relative positioning of skeletal elements of these fossil organisms (line 114-157). We have also substantially revised the Discussion in light of the reviewer’s concerns (line 325-382). We have extensively cited Glass and Blake 2004 throughout the text (e.g., line 126, 129, 132, 164, 168, 208).

6. Reviewer 1 points out that such advances have and are being made with reports of dissociated skeletal parts (e.g. Boczarowski 2001) I reject your response that Isolated ossicles are not available for all taxa (they certainly are), and the identification of isolated ossicles in the absence of complete specimens can be difficult but not impossible and these data can be very informative.

Changes made: This may be the case, but isolated ossicles, by definition, cannot provide information regarding the relative position of skeletal elements. We have clarified this in the text, and emphasized that this information is critical for biomechanical inferences, such as range of motion calculations, and is the basis underlying our choice and advocacy of micro-CT and digital visualization here (line 114-157).

7. As pointed out by Reviewer 2 Line 304: The observation that “there are no extant animals that use two offset columns of repeating segments for musculoskeletally driven locomotion is very interesting. However, I disagree that you can infer that they did not exhibit musculoskeletally-driven motion analogous to that used by extant animals.

Changes made: We have elaborated on the rationale and support underlying the hypothesis that ophiuroids with two offset ambulacrals likely did not engage in musculoskeletally-driven locomotion (lines 325-382). We also provide a framework for further evaluation of this hypothesis (lines 275-278, 374-382).

8. I can see, like Reviewer 1 and Reviewer 2, that there are fundamental problems with conclusion in lines 340 - 342: that is "the expression of musculoskeletally driven locomotion occurred after the evolution of opposing ambulacrals, but preceded the fusion of the ambulacrals to form vertebrae" when you analyse only taxa with ambulacrals either alternating or fused into vertebrae? The key issue is that you have made a broad conclusion based in a limited dataset from just one Lagerstätte the Hunsrück Slate. I do not think you can justify your conclusions by citing Shackleton (2005) as this dataset is Ordovician to Early Silurian or Glass (2005, 2006) which is also restricted to the Hunsrück Slate. In fact, the authors fail to comment on the observation by (Reviewer 2 Line 332). The most important result of this study is the evidence of disparity of locomotory function in the pyritized Hunsrück Slate brittle stars. However, as warning, details of the articular surface of the vertebrae of living brittle stars do not necessarily correspond to differences in locomotory capability. Reviewer 2 points out that there could be a clear there is an evolutionary sequence of development in the arms moving from stepped ambulacrals and inline ambulacrals. However it is almost impossible to identify this trend in the Hunsrück Slate brittle stars due to the limited period of geological time this assemblage covers and the limited dataset (just 5 specimens). Furthermore, such a transformation might have already occurred in the Silurian.

Changes made: We have removed the text referenced by the reviewer from the manuscript. We have reframed the Discussion in light of this concern (line 325-382). We clarify that the 3D images of the best-preserved specimens we examined with two offset rows of ambulacrals

support the previously posited hypotheses that these taxa utilized podial walking for locomotion (line 333-351). We state that more work needs to be done using additional specimens from other localities, taxa, and developmental stages (line 275-278, 374-382). We provide the biomechanical basis as to why musculoskeletally-driven locomotion was likely preceded by the opposition and fusion of ambulacrals to form joint interfaces (line 333-351).

9. I do not see why the “The density of the material provides an appropriate contrast to conduct an exploratory analysis of the advantages of using micro-CT scanning to examine fossil ophiuroid anatomy or the diversity and disparity of ophiuroid taxa within this locality provide an opportunity for a rich comparative analysis”

Changes made: Micro-CT imaging illuminates density differentials within a given specimen. We have now clarified that a high density contrast between the fossil specimens and the slate matrix aids in digital matrix removal with 3D image processing tools (line 168-172). We also note that the Hunsrück Slate has a rich diversity of Palaeozoic ophiuroids (161-164).

10. I reject your conclusion that limiting the investigation to the Hunsrück Slate allows the morphology of each specimen to be directly comparable in terms of their preservation (lines 162-175). Preservation has no influence on the amount of available systematic data. There is amazing brittle star preservation in other localities in the Paleozoic. In order to observe these larger scale trends you need to examine more taxa from more faunas. Its clear that the authors are not listening to the reviewers superior knowledge on any of these points.

Changes made: We have removed the text referenced by the reviewer. A primary aim of this paper was to provide an assessment of the usefulness of micro-CT technology as a tool to visualize the ophiuroids of the Hunsrück Slate (line 156-157). As such, we sought to demonstrate that there is a range of fidelity in terms of preservation even among the better-preserved specimens. We discuss this issue at length in the section on *Material and methods* regarding specimen selection (line 175-186). We also describe the variation in preservation in the specimens selected as the best candidates for micro-CT scanning, providing an honest report of the limitations of this technique as applied to specimens of this locality (e.g., line 175-196, 275-278, 281-283, 296-298, 315-316). We have added additional text clarifying this aim to line 156-157 in the *Introduction*.

11. As per Reviewer 2: Line 357: It’s not correct to say “Prior to this study, the 3D morphology of Palaeozoic brittle star arm ossicles was mainly inferred by integrating the morphology of their dorsal and ventral surfaces exposed by preparation of different specimens.” Such information is already available from many other well-preserved specimens using traditional methods. I don’t any amount of further discussion will hide the fact that all 3 authors do not agree that 3D imaging has revealing previously inaccessible morphological information, such as the articulations between successive segments and the full morphology of a single arm. As we all believe otherwise. This is exemplified by Reviewer 3 Line 361: This is simply not true. The amount of information provided by the CT scanning on the nature, size, shape, and position of the articulation surface between

ambulacrals remains equivocal based on the results presented here. Also, there ARE published accounts of what these articulations looked like in specimens from the groups herein, but not necessarily from the Hunsrück Slate, so CT might not be necessary to provide new insights on new interpretations of these. Even your response that “Articulations” is not synonymous with “articular surfaces” This might be correct, but you can still observe Articulations in disarticulated fossil material.

Changes made: Disarticulated skeletal elements, by definition, are not able to provide information about the position and alignment of successive skeletal structures *in situ*. We agree that much information about Palaeozoic ophiuroid morphology has been gathered with standard methods. However, articulated fossils prepared and visualized with such means do not reveal full 3D geometry and the precise arrangement between successive skeletal structures *in situ* in 360°. These aspects of the skeletal system are necessary for drawing basic biomechanical inferences in organisms that use a musculoskeletal system for locomotion. We clarify this in the Introduction (line 114-157).

12. As sadly is the case with many micro-CT scanning studies. They might show spectacular visual results, but these seldom add to anything new. This MS is still highly speculative and urgently needs additional further functional/experimental analyses. I agree with the associate editor’s previous findings that while the case is made in the manuscript that micro-CT visualization of these fossils reveals new details, that many of these morphological interpretations can be made (and have already been made) using more traditional methods. Its clear from your response to reviews that the authors “fundamentally disagree with the notion that your findings are not novel. I agree this is the first application of the only known technique in which ophiuroid arm morphology can be observed in 360°”. However, these techniques might have spectacular the results, but they do not represent a significant advance in what we already known about these taxa. All three of the previously reviewers agree that micro-CT visualization of an ophiuroids although worthwhile the morphology of these taxa has already been observed 360° using latex casts and disarticulated specimens. The authors have then said this MS is the first to observe ophiuroid arm morphology can in 360°. However, these taxa (Genera) do also exist outside the Hunsrück Slate. The data comes from just only a Devonian single deep-water fauna but then tries to make predictions from taxa (Genera) that have existed from the Ordovician to the Permian (and possibly the Triassic). Although I commend the authors for addressing the recommendations of reviewer 2 and 3. The authors have attempted to add more citations to make up for these failings. It still does not include the vast body of literature that exists on these taxa or knowledge of non-Hunsrück Paleozoic ophiuroids from the Ordovician to the Permo-Triassic.

Changes made: We agree that many of these taxa have been observed in 360° using latex casts and disarticulated specimens. We clarify that the main advantage of using micro-CT in our investigation is the coupled observations of the 3D morphology of individual skeletal elements and their respective positioning *in situ* in 360°. We have added text to line 114-157 in the Introduction and line 194-196 in the Methods noting this and explaining why it is important. The treatment of previous work is appropriate considering our focus was limited to the morphology of specimens described here.

13. Finally, the apparent ignorance of ophiuroid systematics is demonstrated above and by the following comment “We are not familiar with any evidence that “many parts of the ophiuroid skeleton are decalcified,” and are not aware of any non-calcareous skeletal tissue in extant ophiuroids.” Paleozoic ophiuroids do in fact have areas of the skeleton which widely thought to be decalcified.

Changes made: We acknowledge that specimens of Palaeozoic ophiuroids may include incompletely preserved elements due to decalcification or, in the case of Hunsrück specimens, incomplete pyritization. We have considered taphonomic issues in our descriptions of the specimens we scanned by describing the mode of preservation of the skeletal elements and their fidelity, and have taken this into account during specimen selection and functional interpretations (line 175-196, 275-278, 281-283, 296-298, 315-316). We have described measures used to minimize the effects of these issues (lines 175-196, 275-278, 377-381).

14. When reading the revised version, I find much of the discussion is highly speculative as the micro CT can only observed the anatomy and does not test the form and function. This was the real achievement of Clark et al. 2018 looking at analytical functional analyses of the morphology. Replicating the methods from this MS would remove all the highly speculative arguments in this paper regarding arm function and mobility and would better reflect the skills are expertise of the first author.

Changes made: Our reframed Discussion outlines the supporting evidence from our analysis and that from previous reports regarding the development of musculoskeletally-driven locomotion in modern ophiuroids and frames an approach for further evaluation (line 325-382).

Reviewer 4

Reviewer 4 asked that we edit certain sections of the text for clarity, expand on the discussion of *Ophiurina lymani*, and suggest other areas of interest of ophiuroid morphology for further exploration with micro-CT imaging.

1. Line 36 suggest -- four plesiomorphic

Line 38 suggest -- two separately apomorphic

Line 56 suggest -- Apomorphic living ophiuroids

Line 57 suggest -- apomorphic dorsal, ventral

Line 59 suggest -- change ‘Many’ to Plesiomorphic Palaeozoic ophiuroids

Line 139 suggest -- (alternating, plesiomorphic, Lysophiurina)

Line 140 suggest -- (opposing, apomorphic, Zeugophiurina)

Changes not made (no changes necessary): We prefer not to designate the relative origins of these characteristics explicitly as the phylogeny of ophiuroids was not our focus here, nor is there a comprehensive phylogeny of ophiuroids available in the published literature with all taxa

examined here. We have referred to phylogenies of Palaeozoic ophiuroids through the text (e.g., line 76, 81, 125, 188, 189).

2. Line 42 suggest -- This 3D micro-CT scan approach

Changes made: We have added “the use of 3D digital visualization” to line 40.

3. Line 60 suggest -- delete “...in place of.....ventral plates.”

Changes made: We have deleted this section of text as per the reviewer’s suggestion.

4. Line 91 suggest -- add ref 16 -- [16-19, 21]

Changes made: We have added these references (line 73).

5. Line 139 comment -- alternating in conformance with Lovén’s law -- no action required

Changes not made (no changes necessary): Informative, thank you!

6. Line 140 -- please reveal if this is supported by you -- or do you think it is controversial

Changes made: The content of our paper doesn’t provide evidence to support either hypothesis. However, we have changed the syntax of this sentence to alleviate the unintended skepticism conveyed by our phrasing (line 123-125).

7. Lines 264-268 -- your arrows in your Fig. 2 do not match up with placement of groove spines in WK Spencer, his clear text-fig. 262, and his verbal assurance text at bottom of page 408. My experience supports WKS placement. Groove is wide open. Please review. Fix as needed.

Changes made: We have removed the text referencing the groove spines and the arrow in the figure due to uncertainty that they are represented by the pyritized features here.

8. Lines 279 – 298. Is it within scope to make any comparisons here [or in discussion] with ref 46 -- wherein are observations on the hemi-cylindrical ambulacral ossicles of Hunsrück Loriolaster. ...?

Changes made: Several of the observations of the specimens made here are similar to those made in [44]. We have added references to this paper in the description of this taxon (line 281).

9. Lines 300 – 311. Is it within scope to make any comparisons here [or in discussion] with ref 16 -- use *Eospondylus* as proxy for *Furcaster* . Of note is the arrangement in *Furcaster* and in *Eospondylus* of the canal for the radial water vessel being located in the center of the zygosphene knob and the zygotreme pit, like spout and funnel.

Changes made: *Eospondylus* would be an excellent taxon to be analyzed with this method. We have added text suggesting this along with the reference (line 379).

10. Lines 359 - 369 -- I support this interpretation of *Ophiurina vis-à-vis Amphicutus*. Please reference Südkamp's 2017 book *Leben im Devon/Life in the Devonian*, page 143 where this view is mentioned. The similarity of *Ophiurina* and *Amphicutus* is pretty remarkable.

Changes made: We have made the changes suggested by the reviewer (line 366, 656), adding a citation to Südkamp's book.

11. Line 367 -- For myself, the *Ophiomusaidae* and *Ophiosphalmidae* seem very similar to each other, and not so similar at all to *Ophiurina*. You say “with arm morphologies similar to that of *O. lymani*” [as in *Ophiurina lymani* ... yes?]. Is this coming from ref. 60 ...or your analysis?

Changes made: We have changed the text to read: “Living ophiuroids, such as *Ophiomusaidae* and *Ophiosphalmidae* [61,62], with arm structures that show similarities to that of *O. lymani*, use musculoskeletally-driven locomotion.”

12. Line 397 -- add here ----- information comes also from stereo-pair SEM photos of isolated arm ossicles [16, 21]. Stereo-pair images can be created from the 3D micro-CT files as well --- yes?

Changes made: We have added “isolated arm ossicles” to the text here (line 386). Stereo-pair images can be created from 3D micro-CT files to show a snapshot of the whole 3D image: we have made the entire 3D dataset downloadable from FigShare to be included alongside the published manuscript.

13. Line ??? -- 3D micro-CT search for objects of special interest would be of special interest ! Find the madreporite; the madreporite enables ray identification; collect observations related to Loven's law; find the terminal plates. Keep up the search for a periproct. ...

Changes made: We have added: “Micro-CT imaging has the potential to be advantageous for ongoing visualization of the skeletal elements in the disk.” (line 391-392).

Reviewer 5

Reviewer 5 made suggestions regarding terminology and encouraged the use of a range of specimens and developmental stages to limit the influence of taphonomy in morphological interpretations. They also suggested that we use the data from our analysis to evaluate previously posited hypotheses regarding ophiuroid locomotion.

1) Terminology remains unconventional – 1) ‘central axis of the arm’ is the ‘perradial line’ as used by Schuchert and explained clearly in his glossary.

Changes made: We have made the change suggested by the reviewer (line 121).

2) Viewed from the ventral side the ambulacrals are subquadrate to subrectangular. Glass is in error here – the boot-shape he attributes to the ambulacrals refers to ridges on the ambulacrals not to the overall shape of the ossicle.

Changes made: We have clarified this by, for instance, changing the description of the ventral surface as boot-shaped to “the ambulacrals have a boot-shaped ridge on the ventral surface.” (line 126, 217, 241).

3) ‘Row’ for transverse set of at least 2 Ambb + 2 Adambb is preferable to ‘segment’ which has arthropod overtones. There are others remaining.

Changes made: “Segment” is the accepted terminology for the repeating modular series of arm ossicles found in living ophiuroids (e.g., LeClair et al. 1996). We have changed “segment” to “rows of ossicles” when referring to Palaeozoic ophiuroids (e.g., line 185, 200, 316-321, 408).

4) Taphonomic processes. You acknowledge [line241] separation of Ambb columns along the perradial line but as far as I can see do not acknowledge it anywhere else. Separation of Ambb along the columns in *E. tischbeinianus* is surely taphonomic if you compare with the Sth African specimens of this species Ambb columns figured in Jell & Theron. One reviewer points out that taphonomic processes have affected most Hunsruck specimens and he is correct. Your CT scans show much more dislocation than you have explained. It is unreasonable to suggest that the Ambb in your Fig 2b are ‘in situ’ if that means in life position; the gaps between them are undoubtedly taphonomic. Compare the attitude of the Adambb to the perradial line on either side of the arm in your Fig. 2a and you can clearly see evidence of taphonomic disturbance – this is commonplace across stelleroids preserved in any manner and illustrated in any manner.

Changes made: We agree that taphonomic artifacts may have affected the *in situ* position of the skeletal elements. We have walked back the respective conclusions in our revised discussion (lines 325-382) and outlined directions for future analysis of additional specimens to minimize the influence of taphonomy on the interpretation of the *in situ* positioning of the ossicles (lines 275-278, 377-380, 399-401). We also describe the state of preservation of the specimen used in this analysis (e.g., line 175-196, 281-283, 296-298, 315-316).

3) Growth. From the scales given in your Fig. 2a vs 2b the arm of *E. tischbeinianus* (2b) is somewhere near twice the size of the arm of *E. roemeri* (2a). They clearly represent different growth stages because Lehmann 1957 illustrated similarly sized specimens of these two species; they therefore might be expected to be different. For an accurate comparison, adult specimens of the two species ought to be compared.

Changes made: Quality of internal preservation was the main driver in specimen selection (see lines 175-196). We agree that the broader application of our method to specimens of increasing size is necessary to explore the development of the 3D geometry of the arm ossicles and their relative positioning. We have added text stating this (line 275-278, 377-380, 399-401).

4) I agree with the earlier reviewer who pointed out that your whole discussion of locomotion in these animals is previously understood and published interpretation and as far as I can see that reviewer is correct in asserting that the CT scans have no bearing on that discussion. I suggest you write a review of ophiuroid locomotion without reference to your scans and then when you have finished look to see if the scans add anything extra. That is to say instead of starting with the scanning and assuming it tells you more than was previously known a priori, see what is known then see if the scanning adds anything.

Changes made: Please see our response to Comments 1 and 2 by the Associate Editor. We have added text regarding the importance and novelty of our new data (lines 114-157) and the potential of this approach to make critical contributions to inferences regarding the locomotion strategies of fossil ophiuroids (line 325-382). This addresses the suggestions of the reviewer in a way that we feel best matches the goals of this study.

5) It appears that the initial interest was to see what CT scanning of these fossilised animals would produce in relation to earlier CT scanning of living ophiuroids and the various questions surrounding Palaeozoic ophiuroids has been investigated superficially as noted by two earlier reviewers who sought more integration with the considerable body of knowledge already available on Palaeozoic ophiuroids and largely ignored herein e.g. Spencer provided excellent illustrations of the articulating proximal and distal faces of *Ambb* in *Encrinaster grayae* but you fail to mention these or compare with *E. roemeri*. I could go on but the list is quite long.

Changes made: In the previous versions of the text, we failed to clarify the distinction between “articulations” and “articular surfaces.” By “articulations,” we mean the 3D position of the skeletal elements with respect to one another, which was previously unknown. This is in contrast to the nature of the surfaces on the elements where they articulate (i.e. “articular surfaces”), which has been documented in the literature. We clarify this in this latest version of the text (line 136-149, 194-195, 374-376, 387-390), and elaborate that micro-CT scanning is the only non-destructive method to visualize the 3D morphology and their relative positioning with respect to one another *in situ* (line 114-157). The illustrations of *Encrinaster grayae* noted by the reviewer are of the oral surfaces, not the articulating faces.

6) Your assertion that *Encrinaster* and *Euzonosoma* are not synonymous in line with Blake and Glass makes the same mistake as they did in canvassing only the two Hunsruck species (Spencer did the same thing in erecting *Euzonosoma* when he compared the two Scottish species). You need to canvas the entire species content and I suggest you will find a full gradation in the length of arms, number of rows per arm, arm width etc. Only after such an examination ought you to make a decision.

Changes made: Our results provide further evidence to support the distinction between *Encrinaster* and *Euzonosoma*. As the reviewer states, we agree that if we examine more specimens we should find differences in the length of arms, number of rows per arm, arm width, etc. However, the synonymization of the two genera made by Jell and Theron was based on a different facet of their anatomy: the mediolateral width of the ambulacrals. Jell and Theron argued that it was a taphonomic process that shifted the angle at which these ossicles were exposed, making it appear as though their dimensions varied when they were the same in reality. Our investigation revealed these ossicles *in situ* in 3D, demonstrating major shape differences between them that cannot be attributed to their rotation. Thus, we felt it important to discuss this evidence. We have added a suggestion that further 3D imaging should be performed to reveal whether taphonomic processes shifted the angle at which these ossicles were exposed, or if major shape differences are prevalent across taxa (line 275-278).